# Analysis of lung transcriptome in calves infected with Bovine Respiratory Syncytial Virus and treated with antiviral and/or cyclooxygenase inhibitor

Maxim Lebedev[1], Heather A. McEligot[1], Victoria N. Mutua[1], Paul Walsh[2], Francisco R. Carvallo Chaigneau[3], Laurel J. Gershwin[1]*

1 Department of Pathology, Microbiology and Immunology, School of Veterinary Medicine, University of California Davis, Davis, California, United States of America, 2 Pediatric Emergency Medicine, Sutter Medical Center Sacramento, Sacramento, California, United States of America, 3 Department of Biomedical Sciences & Pathobiology, Virginia-Maryland College of Veterinary Medicine, Virginia Tech., Blacksburg, VA, United States of America

* ljgershwin@ucdavis.edu

**Data Availability Statement:** All sequence read files are available from the NCBI SRA database (accession number PRJNA663348).

## Abstract

Bovine Respiratory Syncytial virus (BRSV) is one of the major infectious agents in the etiology of the bovine respiratory disease complex. BRSV causes a respiratory syndrome in calves, which is associated with severe bronchiolitis. In this study we describe the effect of treatment with antiviral fusion protein inhibitor (FPI) and ibuprofen, on gene expression in lung tissue of calves infected with BRSV. Calves infected with BRSV are an excellent model of human RSV in infants: we hypothesized that FPI in combination with ibuprofen would provide the best therapeutic intervention for both species. The following experimental treatment groups of BRSV infected calves were used: 1) ibuprofen day 3–10, 2) ibuprofen day 5–10, 3) placebo, 4) FPI day 5–10, 5) FPI and ibuprofen day 5–10, 6) FPI and ibuprofen day 3–10. All calves were infected with BRSV on day 0. Daily clinical evaluation with monitoring of virus shedding by qRT-PCR was conducted. On day10 lung tissue with lesions (LL) and non-lesional (LN) was collected at necropsy, total RNA extracted, and RNA sequencing performed. Differential gene expression analysis was conducted with Gene ontology (GO) and KEGG pathway enrichment analysis. The most significant differential gene expression in BRSV infected lung tissues was observed in the comparison of LL with LN; oxidative stress and cell damage was especially noticeable. Innate and adaptive immune functions were reduced in LL. As expected, combined treatment with FPI and Ibuprofen, when started early, made the most difference in gene expression patterns in comparison with placebo, especially in pathways related to the innate and adaptive immune response in both LL and LN. Ibuprofen, when used alone, negatively affected the antiviral response and caused higher virus loads as shown by increased viral shedding. In contrast, when used with FPI Ibuprofen enhanced the specific antiviral effect of FPI, due to its ability to reduce the damaging effect of prostanoids and oxidative stress.

**Funding:** This work was funded by USDA NIFA (dual purpose/dual benefit) grant #2017-67015-26083 awarded to LJG and PW (http://www.nifa.usda.gov) and UC Davis Comparative Medical Science Training Program (NIH Grant #T32 OD011147) awarded to Nicole Baumgarth, which funded trainee VNM (https://www.nih.gov). The DNA Technologies and Expression Analysis Cores at the UC Davis Genome Center are supported by a NIH shared Instrumentation Grant 1S10OD010786-01. The funders had no role in study design, data collection and analysis, decision to publish, or preparation of the manuscript.

**Competing interests:** The authors have declared that no competing interests exist.

## Introduction

Bovine respiratory disease complex (BRDC) is a common and serious illness of both dairy and beef cattle. This multi-factorial and multi-pathogenic condition [1] is a leading cause of direct and indirect economic losses in the dairy- and especially in the beef industry [2, 3]. One of the most common pathogens in the etiological structure of BRDC is bovine respiratory syncytial virus (BRSV) [4]. BRSV belongs to genus *Orthopneumovirus*, family *Pneumoviridae* (https://talk.ictvonline.org/taxonomy/). It is a single-stranded negative-sense RNA virus with the high tropism to epithelial cells of the upper- and lower respiratory tract. Clinical manifestation of the disease is characterized by rhinitis, laryngopharengitis, tracheo-bronchitis, and, in more severe form, bronchiolitis with wheezing, tachypnea and cough. Clinical signs in calves include: fever, anorexia, and depression.

Human respiratory syncytial virus (RSV) infection, caused by human orthopneumovirus, has virtually the same symptoms and pathogenesis as bovine respiratory syncytial virus infection in calves. Based on the similarity of the disease and genetic and antigenic similarity between bovine RSV and human RSV, bovine calves are considered an excellent model for experimentation and trials as the results are easy to extrapolate to human infants [5]. Bronchiolitis in infants is the major cause of hospitalizations and morbidity in the United States and other countries [6–8].

It has been demonstrated that RSV infection increases expression of cyclooxygenase-2 (COX-2) and prostaglandin E2 (PGE2) production in human alveolar epithelial cells. In a cotton rat (*Sigmodon hispidus*) model RSV increased expression of cyclooxygenase-2 (COX-2) with the peak on day 5 [9]. Similarly, BRSV can increase COX-2 cellular expression in airway bronchiolar and bronchial epithelial cells and macrophages of neonatal lambs (*Ovis aries*) [10]. Increase of PGE2 and thromboxane B2 concentrations were reported in plasma and lung lavage of calves (*Bos taurus*) infected with BRSV [11]. This makes COX-2 a good target for therapeutics, helping to minimize potential damage caused by excessive production of prostanoids and alleviate pathological consequences of BRSV and RSV infection. Nonsteroidal anti-inflammatory drugs (NSAIDs) or COX inhibitors have the potential to diminish the prostanoid surges that follow RSV infection. Pretreatment with NSAIDs or treatment at 24 hours post RSV inoculation decreases the histopathological changes in a cotton rat model [9]. Ibuprofen is one of the widely used COX inhibitors and has been proven to be safe to use in pediatric practice [12–14]. We have previously shown improved clinical scores, but increased viral load when ibuprofen was administered to calves experimentally infected with BRSV [15].

It has been previously demonstrated that NSAIDs help to improve clinical outcomes of specific antiviral agents and combined antiviral and anti-inflammatory therapy of the acute RSV infection looks most promising, as it was previously demonstrated with the combination of anti-RSV antibody therapy with NSAID or corticosteroids [16]. Besides the antibody therapy, the RSV fusion protein inhibitor (FPI) has been recently developed as a specific anti-RSV therapeutic. It blocks virus entry by inhibiting the fusion of the viral envelope with the host cell membrane. The BRSV model was used to evaluate the antiviral efficacy of this therapeutic agent, GS-561937 (or GS1). It is a close structural analog of GS-5806, RSV FPI, developed for humans. GS1 demonstrated clear therapeutic effect by reducing the viral load, disease symptom score and lung pathology in experimental calves [17]. Similarly, in a challenge study of healthy adults, treatment with GS-5806 reduced the viral load and the severity of clinical presentation [18]. These promising results raised a question regarding the efficacy of combined FPI and COX inhibitor therapy and the possible benefit of this combination suggesting that the combination of a NSAID and a specific antiviral agent can improve the outcome of the disease even more than one treatment alone.

We have recently performed an experiment to address this hypothesis and have shown that the combination of the antiviral GS1 FPI and the COX inhibitor ibuprofen significantly reduced both clinical scores and viral loads as measured by viral shedding in experimental BRSV infection of pre-ruminant calves. This effect was most apparent when therapy was initiated on the third day after viral infection [19]. The percent of lung consolidation at necropsy on day 10 after BRSV infection was significantly less in the animals receiving dual treatment beginning on day 3 after infection. The mean lung consolidation scores for treatment groups are presented here in the results section. Histopathological scores for lung sections from each calf were tabulated and also revealed a significant treatment effect for dual therapy beginning on day 3 post infection. Administration of ibuprofen alone increased viral loads, but combined treatment helped to decrease the viral load even more than in the treatment group which received FPI as the sole therapeutic, suggesting that these two therapeutics have a synergistic effect.

Comparative functional analysis of the lung tissue in different treatment groups, using gene expression analysis, can give a detailed picture of changes caused by different treatments. Previous transcriptomics study of the lung tissue at the peak of BRSV infection without treatment predicted high production of nitric oxide and reactive oxygen species activation, lung tissue damage, complement and coagulation cascades, endocytosis, chemokine and cytokine signaling, leukocyte transendothelial migration, cell adhesion and MAPK signaling. These functions demonstrated a coordinated immune defense that occurs in the lung tissue to combat the infection [20]. Herein we describe results of the analysis of the transcriptome in lung tissue samples, collected at necropsy on day 10 after infection and 7 or 5 days of treatment. We analyzed the influence of potential treatment options with FPI and ibuprofen on the lung tissue of BRSV-infected calves. We have explored how treatments change transcriptomes of lung areas with gross pathological changes when compared with areas of the lung without visible changes. We also compared transcriptomes of both types of the lung tissue between the placebo group (placebo treatment but infected) and all treatment groups, which included treatment with either FPI alone or ibuprofen alone [19].

## Materials and methods

### Animal procedures

The study was performed using bovine calves (*Bos taurus*) and approved by the University of California Davis Institutional Animal Care and Use Committee (authorization number 19313). Detailed description of the study design, randomization, interventions, outcomes, handing and detection of the virus has been provided elsewhere [19]. Briefly, 36 healthy five to six-week-old outbred pre-ruminant bottle-fed Holstein bull calves were randomly split into 6 treatment groups. All animals were infected with BRSV by aerosolization of 5 ml of infected bovine turbinate cell culture supernatant, using individual nebulizers, connected to a tightly fitted face respirator mask. Infection dose of the virus was $3.9 \times 10^5$–$9.7 \times 10^5$ PFU. The following treatment groups were created as follows: Group 1 –ibuprofen, starting on day 3 post infection; Group 2 –ibuprofen, starting on day 5 post infection; Group– 3, placebo; Group 4 –the antiviral fusion protein inhibitor (FPI), starting on day 3 post infection; Group 5 –FPI and ibuprofen starting on day 5 post infection; Group 6 –FPI and ibuprofen starting on day 3 post infection. GS-561937 (FPI) was administered 600mg per animal in 30 ml of a mixture of propylene glycol and First Street Snow Cone Syrup (Amerifoods Inc, Los Angeles, CA) once daily. Ibuprofen was administered at 10 mg/kg, three times daily, in the form of Advil suspension for children (100 mg/5ml—Pfizer inc, Madison, NJ). The First Street Snow Cone Syrup without ibuprofen was administered as an ibuprofen placebo, as well as its 50:50 mix with propylene

glycol without FPI for those groups that did not receive FPI (placebo). All therapeutics, including placebo, were given orally, using catheter tipped syringes.

Clinical scores as one of the outcome measurements were used as described elsewhere [21]. Nasal swabs were collected daily for RNA extraction and qRT-PCR was performed to monitor virus shedding as described [19]. On day 10 post infection all animals were euthanized with an intravenous injection of sodium pentobarbital and necropsy was performed.

## Microbiology culture of tissues at necropsy

For each animal a swab was taken from a mainstem bronchus and one was taken from lung, by searing the outside of the tissue and inserting a sterile swab into the tissue beneath. The swab was rotated in the tissue and then carefully placed in a sterile container with transport media for culture. The samples were inoculated onto diagnostic media at the California Animal Health and Food Safety Diagnostic Laboratory at the University of California, Davis. Samples had aerobic bacterial culture performed and results were reported after 48 hours of incubation.

## Tissue samples collection and RNA extraction

Lung tissue was collected at necropsy on day 10 after infection. Two types of lung tissue samples were collected as small pieces, immediately flash-frozen in liquid nitrogen and stored at -80˚C. Lung tissue samples of one type were taken from areas with gross pathological changes or visually observed lesions (Fig 1) and is further mentioned here as "lung lesion" (LL). Lung tissue samples of the second type from the same animal were collected from the area of the lung without visible pathological changes and labeled as "lung non-lesional" (LN).

A thin slice (30mg) of each frozen lung tissue sample was taken using a razor blade and immediately placed into 0.6 ml of buffer RLT (Qiagen, Germany) on the surface of a sterile Petri dish to start lysis. Once in the lysis buffer, the piece of tissue was chopped with the razor blade to multiple smaller pieces and placed into a 1.5 ml tube (Eppendorf) and vortexed thoroughly to help homogenization. In order to reach the finest homogenization, samples were centrifuged through the QIAshredder spin column (Qiagen). Centrifugate was diluted 1/1 with 70% alcohol and RNA extracted using RNeasy Mini Kit (Qiagen) according to the manufacturer's instructions. To eliminate residual genomic DNA contamination, on-column DNAse digestion was performed using RNAse-Free DNAse Set (Qiagen), according to the manufacturer's instructions. After elution, RNA was checked for RNA concentration and contamination using NanoDrop 2000c spectrophotometer (Thermo Scientific, USA) and tested for RNA integrity, using Bioanalyzer 2100 (Agilent, Santa Clara, CA USA) according to manufacturers' instructions. Extracted and quality checked RNA was stored at -80˚C until used in further procedures.

## RNA sequencing

Library generation and sequencing was performed in the DNA technology & Expression Analysis Core Laboratory at the University of California Davis. Barcoded 3' Tag-Seq libraries were prepared using the QuantSeq FWD kit (Lexogen, Vienna, Austria) for multiplexed sequencing according to the recommendations of the manufacturer. The fragment size distribution of the libraries was verified via micro-capillary gel-electrophoresis on a Bioanalyzer 2100 (Agilent, Santa Clara, CA). The libraries were quantified by fluorometry on a Qubit instrument (Life-Technologies, Carlsbad, CA), and pooled in equimolar ratios. Forty eight libraries were sequenced per lane on a HiSeq 4000 sequencer (Illumina, San Diego, CA) with single-end 100 bp reads. The sequencing generated more than 3 million reads per library.

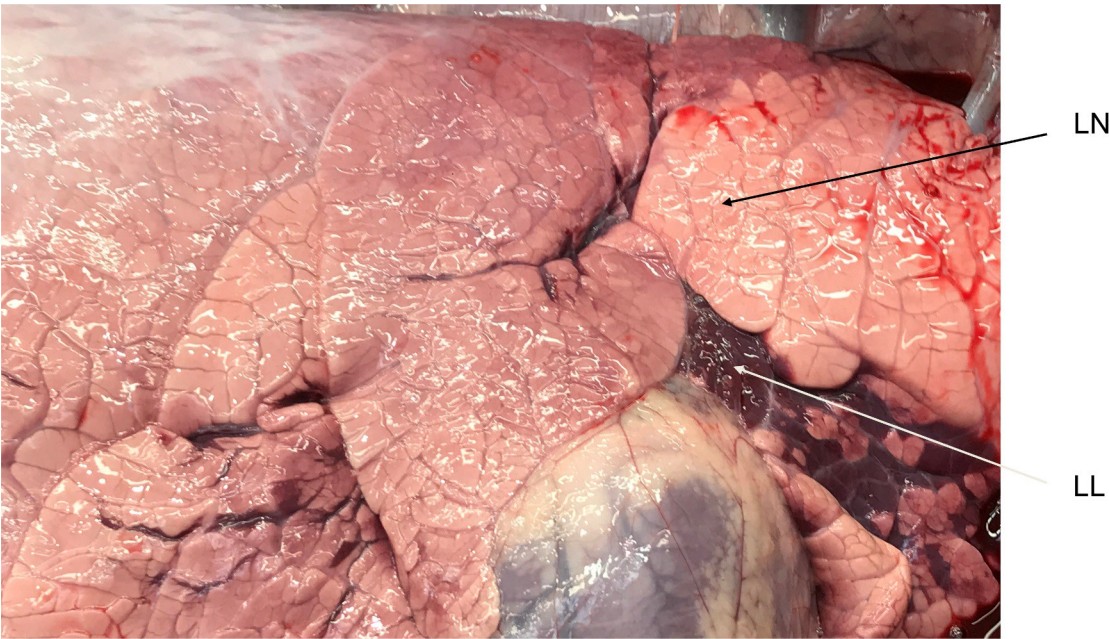

**Fig 1. Example of lung areas representing "lung lesion" (LL) and "lung non-lesional" (LN).** Lung tissues for gene expression analysis were sampled from areas with gross pathological changes (lesions) and areas where pathological changes were not visually observed.

## Analysis of RNA sequencing data

Analysis of the sequencing data and differential gene expression was performed by UC Davis Bioinformatics Core. Differential expression analyses were conducted using limma-voom [22]. The model fitted included effects for sample type, treatment, the interaction between sample type and treatment, sequencing pool, and RNA isolation date. Standard errors of log fold changes were adjusted for within-animal correlations. Gene ontology enrichment analysis of the whole transcriptome was conducted using Kolmogorov-Smirnov test as implemented in the Bioconductor package topGO, version 2.30.1 [23]. KEGG pathway [24] enrichment analysis was conducted using Wilcoxon rank-sum tests, with KEGG pathway annotation obtained using the Bioconductor package KEGGREST, version 1.18.1 (https://www.bioconductor.org/packages/release/bioc/manuals/KEGGREST/man/KEGGREST.pdf). *Bos taurus* Ensembl gene identifiers and annotations were used in this study [25].

## Functional analysis and interpretation

Area-proportional Venn diagrams and analysis of differentially expressed gene lists were performed using BioVenn [26]. Functional enrichment analysis of differentially expressed genes and visualization of functional networks was performed using Cytoscape/ClueGo version 2.5.7 [27]. Only genes with $p < 0.05$ were included into the analysis. Functional enrichment analysis was conducted using a two-sided hypergeometric test with Bonferroni step down correction. Latest "GO Bological Process", "GO Immune System Process" and KEGG databases were used. GO levels were set in the interval from 3 to 8, minimum of genes per GO term or KEGG pathway threshold was 2 genes and 4%. The Kappa score threshold of 0.4 was set to determine term-term interactions and visualization of term connections as a network. ClueGo 2.5.7 was also used for visualization of networks, comparative analysis and sorting of whole

transcriptome enrichment results using predefined terms from topGO and KEGGREST analyses. Only terms and pathways with p<0.05 were included to the input data.

## Results

### Differential gene expression in lung lesions in comparison with non-lesion lung tissue

Differential gene expression analysis was performed in the comparison between affected lung tissue with visible morphological changes–lung lesion (LL) and lung tissue without lesion– lung normal (LN). After filtering out genes with fewer than 4 counts per million, 8171 genes were remaining for the analysis.

Multidimensional scaling (MDS) plots (S1 Fig) did not show strong separation of samples from any treatment group but demonstrated apparent difference between samples from lesions and lung tissue without lesions. The highest number of differentially expressed genes (60 genes) with p<0.05 was observed in group 3 –placebo (Table 1), the lowest number of differentially expressed genes was in group 1, treated with only ibuprofen, starting at day 3. In other groups the number of differentially expressed genes was in the range from 36 to 48. Lists of genes are presented in the S1 File). Area-proportional Venn diagrams (Fig 2) visually demonstrate that the number of differentially expressed genes was relatively smaller in groups where ibuprofen was administered without FPI in comparison to the placebo group, which showed 39.39% differentially expressed genes that were unique for this group in this set of comparisons. A similar picture was observed when these two groups were compared to group 4, treated with fusion protein inhibitor only. Groups with combined (ibuprofen + FPI) treatment demonstrated larger proportions of non-shared DE genes and, at the same time, larger proportions of genes shared with both placebo and FPI only groups. In the attempt to identify functional significance of these differences in gene expression, lists of genes that differentially expressed uniquely in certain treatment groups were generated (S2 File). Comparing gene IDs of 3 lists of genes uniquely expressed in sets of comparison of groups 1-2-3, 1-3-6 and 2-3-5 demonstrated 37.5% (data not shown) genes common for all 3 sets and genes from 2-3-5 almost completely overlap with 1-2-3. Functional analysis of these 3 gene sets demonstrated following GO terms significantly enriched in all 3 gene sets: "negative regulation of transcription from RNA polymerase II promoter in response to stress" (adjusted p = 0.0003), "regulation of transcription from RNA polymerase II promoter in response to stress" (p = 0.003) and "regulation of DNA-templated transcription in response to stress" (p = 0.002). 2 genes (DNAJB1 and TMBIM6) are associated with these GO terms and were uniquely differentially expressed in lung lesions of the placebo group. Notably, expression of DNAJB1 was 3.15 log fold change (logFc) higher in lung lesion than in non-lesion, but TMBIM6 was 0.8 logFc lower in lung lesion. Both genes had relatively high average expression, if compared to other genes.

**Table 1. Number of differentially expressed genes in the comparison of lung lesion tissue vs tissue from same lung without lesions for each treatment.**

| Treatment group | Differentially expressed genes (with adjusted P < 0.05) |
|---|---|
| 1 –Ibuprofen, day 3–10 | 29 |
| 2 –Ibuprofen, day 5–10 | 36 |
| 3 –Placebo | 60 |
| 4 –FPI, day 5–10 | 40 |
| 5 –FPI+Ibuprofen, day 5–10 | 48 |
| 6 –FPI+Ibuprofen, day 3–10 | 41 |

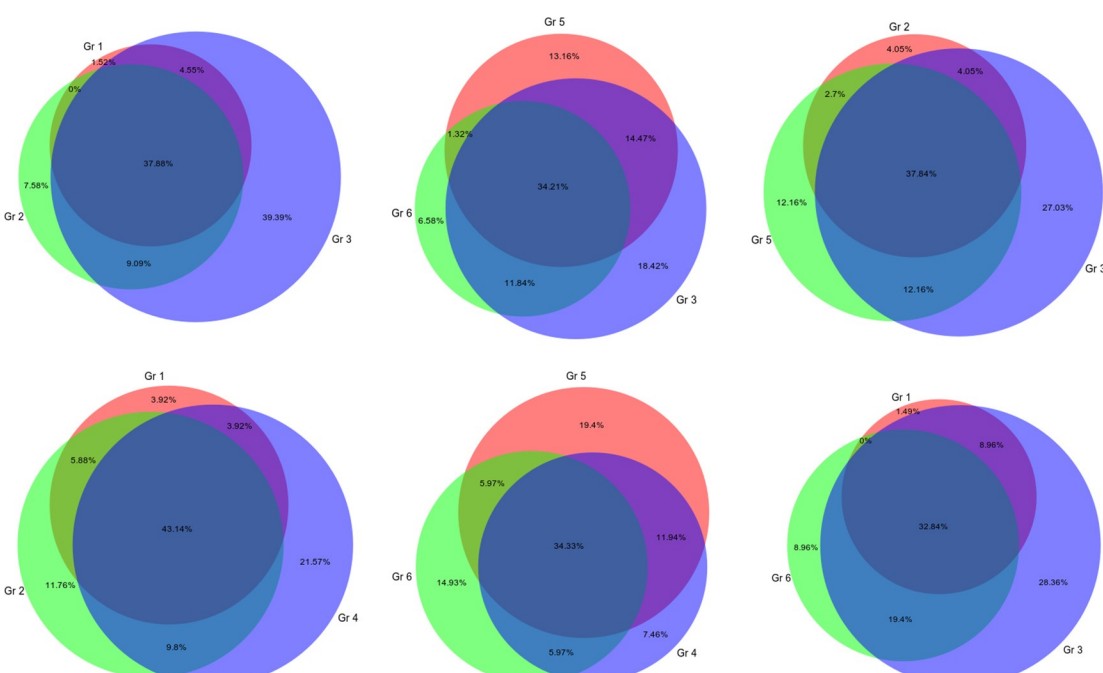

**Fig 2. Venn diagrams show % of differentially expressed genes in lesional lung tissue (LL) compared to lung tissue without gross lesions (LN).** Each circle represents a list of differentially expressed genes (p<0.05) when LL is compared to LN within a treatment group.

The last pair of Venn diagrams on Fig 1 show comparison of differential gene expression in groups treated with ibuprofen only or a combination of FPI and ibuprofen, both started at day 5 after infection; the same combination of treatments, but started at day 3, is also compared to the placebo. It is of interest to see how gene expression differs in lung lesions when combined treatment was administered, since it was the most clinically effective combination. Lists of unique genes that were differentially expressed in lung lesions only in groups with combined treatment shown in S2 File. The gene lists are relatively short and functional analysis (with threshold of 2 genes per GO term or KEGG pathway) demonstrated only one GO term enriched in group 5; it was "positive regulation of potassium ion transport" (p = 0.00011). 2 DE genes (FHL1, WNK1) were associated with this term. Expression of FHL1 was 1.85 logFc lower and WNK1 was 1.07 lower in lung lesion, with relatively high average expression of these genes. Even though functional analysis did not bring any enriched pathways for group 6, differential expression of CD180 and CMTM7 deserves close attention from an immunological stand point. Expression of CD180 was 3.9 logFc lower and CMTM7 was 2.1 logFc lower in lung lesion than in non lesion lung.

Gene ontology (GO) enrichment analysis demonstrated that these differentially expressed genes are associated with a total of 33 GO terms, among them, there were 24 GO terms with adjusted p<0.05 (Table 2). 10 out of 24 significantly enriched terms were related to the immune system functions, such as innate immune functions, including activation of neutrophils and the classical pathway of complement activation. Adaptive immune functions included: B cell differentiation and activation, antigen recognition, and humoral immune response mediated by circulating immunoglobulin. However, none of the immune system related terms were specific to a certain treatment group. granulocyte/neutrophil activation GO terms (GO:0042119, GO:0036230) were not enriched in samples from group 6 (day 3–10 FPI and ibuprofen). In other treatment groups, the ANXA3 (annexin A3) and/or FCGR3A genes

**Table 2. GO enrichment analysis results of differentially expressed genes in lesions versus non-lesion lung tissue.**

| GO/KEGG ID | GO Term | Term P Value (adjusted) | Treatment group |
|---|---|---|---|
| GO:0002335 | mature B cell differentiation | 0.00131 | No Specific Gr |
| GO:0046110 | xanthine metabolic process | 0.00417 | No Specific Gr |
| GO:0098883 | synapse pruning | 0.00577 | No Specific Gr |
| GO:0045601 | regulation of endothelial cell differentiation | 0.00585 | No Specific Gr |
| GO:0002312 | B cell activation involved in immune response | 0.00934 | No Specific Gr |
| GO:0150146 | cell junction disassembly | 0.00987 | No Specific Gr |
| GO:0030858 | positive regulation of epithelial cell differentiation | 0.01686 | No Specific Gr |
| GO:0097201 | negative regulation of transcription from RNA polymerase II promoter in response to stress | 0.01840 | Specific for Placebo |
| GO:0005044 | scavenger receptor activity | 0.02033 | No Specific Gr |
| GO:0002313 | mature B cell differentiation involved in immune response | 0.02360 | No Specific Gr |
| GO:0045603 | positive regulation of endothelial cell differentiation | 0.03242 | No Specific Gr |
| GO:0043620 | regulation of DNA-templated transcription in response to stress | 0.03246 | Specific for Placebo |
| GO:0003158 | endothelium development | 0.03312 | No Specific Gr |
| GO:0042119 | neutrophil activation | 0.03821 | No Specific Gr |
| GO:0038024 | cargo receptor activity | 0.03828 | No Specific Gr |
| GO:0050857 | positive regulation of antigen receptor-mediated signaling pathway | 0.03884 | No Specific Gr |
| GO:0006958 | complement activation, classical pathway | 0.03952 | No Specific Gr |
| GO:0043618 | regulation of transcription from RNA polymerase II promoter in response to stress | 0.04140 | Specific for Placebo |
| GO:0050854 | regulation of antigen receptor-mediated signaling pathway | 0.04140 | No Specific Gr |
| GO:0036230 | granulocyte activation | 0.04439 | No Specific Gr |
| GO:0030301 | cholesterol transport | 0.04838 | No Specific Gr |
| GO:0001974 | blood vessel remodeling | 0.04871 | No Specific Gr |
| GO:0002455 | humoral immune response mediated by circulating immunoglobulin | 0.04871 | No Specific Gr |
| GO:0006144 | purine nucleobase metabolic process | 0.04871 | No Specific Gr |
| KEGG:00500 | starch and sucrose metabolism | 0.01552 | No Specific Gr |
| KEGG:05340 | primary immunodeficiency | 0.01312 | No Specific Gr |
| KEGG:04610 | complement and coagulation cascades | 0.00172 | No Specific Gr |
| KEGG:05020 | prion diseases | 0.01552 | No Specific Gr |

from granulocyte/neutrophil activation GO terms were differentially expressed; in all groups these genes had significantly lower levels of expression.

There are another 3 overlapping GO terms (GO:0097201, GO:0043620 and GO:0043618) that are not directly related to the immune system, but may be of great importance for the cell stress markers. All 3 terms are related to regulation of nucleic acid synthesis in response to stress. They are significantly enriched in samples from treatment groups 3 (placebo) and 6 (Day 3–10 FPI and Ibuprofen). DNAJB1 (DnaJ heat shock protein family (Hsp40) member B1) is the gene belonging to these GO terms and was differentially expressed (upregulated) in lesions of both treatment groups. Another gene TMBIM6 (transmembrane BAX inhibitor motif containing 6) was differentially expressed (downregulated) only in lesion samples from group 3.

Generation of a network among GO terms using predicted network interactions (Fig 3) demonstrated close relationships between B cell activation/differentiation terms and antigen receptor-mediated signaling. The close relationship of this functional cluster with purine base metabolism GO terms, points to active nucleic translation and transcription processes. Genes (ADA and MFNG), that are differentially expressed in B cell activation pathways have significantly lower expression in lesions than in the lung tissue without lesion. ADA is the shared gene between these two functional clusters. Another gene (PRKCB) belonging to the antigen receptor-mediated pathway is down-regulated in the lesions too, demonstrating that antigen

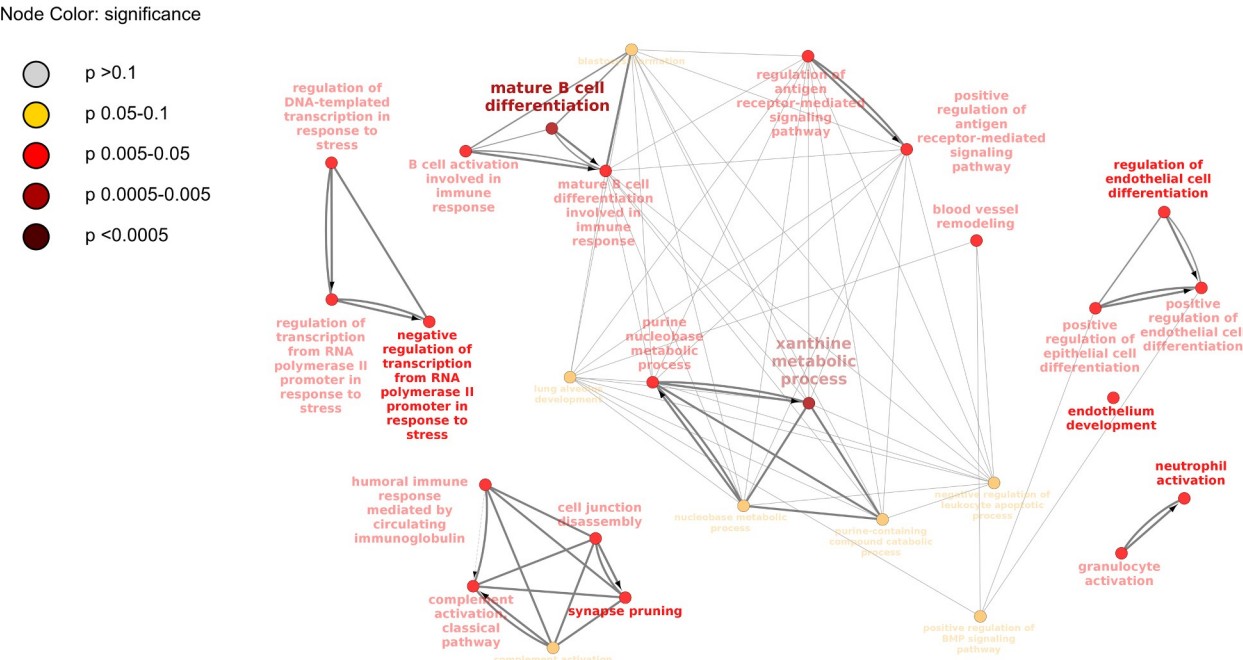

**Fig 3. A map of predicted connections among GO terms based on the analysis of differentially expressed genes in lesional lung tissue (LL) vs lung tissue without lesions (LN) is shown.** Gene ontology enrichment analysis of differentially expressed genes (p<0.05) in each treatment group was conducted and results were visualized as a map of predicted connections among significantly enriched GO terms with Kappa threshold of 0.4.

response, B cell differentiation and proliferation is suppressed in affected lung tissue with apparent pathological changes.

Another functional cluster on Fig 3 is related to the classical complement pathway and represented by C1QA and C1QB genes, with significantly lower expression in the affected lung tissue with lesions, suggesting either suppressed production of C1q complement component in lesions or increased production of classical complement components in lung tissue areas without lesions. Neutrophil activation terms stand alone on the map and have no predicted connections with the two functional clusters, mentioned above.

Another functional group of GO terms is related to vascular remodeling and endothelial cells differentiation and is not directly related to the immune function, but contributes to the lung tissue dynamics in BRSV infected animals. All related genes (CDH5, VEZF1, XDH, BMPR2, SLC40A1, AXL) had significantly lower levels of expression in lung lesion samples.

KEGG pathway enrichment analysis showed 4 pathways significantly enriched when only differentially expressed genes were used as an input data. Two pathways relevant to this study are "primary immunodeficiency" and "complement and coagulation cascades." Genes ADA and CD8A are associated with the primary immunodeficiency and both had reduced expression in lesions of the lung. Complement and coagulation cascades included the following differentially expressed genes: C1QA, C1QB, CR2, PROS1. All these genes had significantly lower levels of expression in lung tissue with lesions.

## GO and KEGG enrichment analysis based on the comparison of whole transcriptomes in the lung tissue with- and without lesions

Additionally, GO enrichment analysis of the whole transcriptome in samples from LL in comparison with LN was performed for all treatment groups. The analysis determined

approximately 50–85 GO terms with p<0.05 per treatment group. Since the goal of this study is to explore how immune mechanisms were modulated by different types of treatment, we have selected GO terms related to immune functions or relevant to some other aspects of this study. Analyzing general trends in expression of genes comprising these GO terms we determined whether this function was relatively increased or decreased in LL compared to LN. The result (Table 3) demonstrated that one of the most active processes in the lesions of the lung was leukocyte migration, especially in groups 1 and 6 in which ibuprofen was administered starting from day 3. But, in group 1, not treated with FPI, this function appeared to be more activated and involved GO terms related to chemotaxis of all major immune cell types (monocytes/macrophages, neutrophils and lymphoid cells). Interestingly, negative regulation of viral genome replication was activated only in groups 1 and 2 (both treated with ibuprofen without FPI), but not in the placebo group 3. Some unique functions in LL from group 3 were increased, including oxidative stress-related terms and regulation of angiogenesis. GO terms enriched in all groups with

**Table 3. Enriched GO terms related to the immune functions in lung lesion tissue.**

| Treatment Group | Increased function | Decreased function |
|---|---|---|
| 1 –Ibuprofen, day 3–10 | monocyte chemotaxis | response to toxic substance |
| | chemokine-mediated signaling pathway | positive regulation of B cell proliferation |
| | macrophage chemotaxis | adaptive immune response |
| | neutrophil chemotaxis | regulation of entry of bacterium into host cell |
| | lymphocyte chemotaxis | positive regulation of interleukin-17 production |
| 2 –Ibuprofen, day 5–10 | defense response to virus | medium-chain fatty acid metabolic process |
| | (RSAD2 (Viperin) increased 2.4 times) | leukocyte tethering or rolling |
| | Chemotaxis | positive regulation of interferon-gamma production |
| 3 –Placebo | regulation of angiogenesis | regulation of protein processing |
| | oxidation-reduction process | |
| | respiratory electron transport chain | |
| | cellular response to oxidative stress | |
| | positive regulation of T cell migration | |
| 4 –FPI, day 5–10 | cytokine-mediated signaling pathway | T cell costimulation |
| | positive regulation of T-helper 2 cell cytokine production | defense response to Gram-positive bacterium |
| 5 –FPI+Ibuprofen, day 5–10 | No terms | positive regulation of interferon-beta production |
| | | negative regulation of cell proliferation |
| | | positive regulation of cytokine production |
| 6 –FPI+Ibuprofen, day 3–10 | induction of positive chemotaxis | type I interferon biosynthetic process |
| | | positive regulation of TOR signaling |
| | | response to type I interferon |
| | | antigen processing and presentation of exogenous peptide antigen via MHC class I |
| Nonspecific terms (Enriched in groups 1 and 2) | negative regulation of viral genome replication | No terms |
| Nonspecific terms (enriched in 2 or more groups) | negative regulation of autophagosome assembly | neutrophil activation |
| | response to antibiotic | cellular response to interleukin-4 |
| | antimicrobial humoral immune response mediated by antimicrobial peptide | cytokine production |
| | prostaglandin biosynthetic process | |
| | antibacterial humoral response | |
| | positive regulation of toll-like receptor 3 signaling pathway | |

increased function were toll-like receptor 3 signaling, antimicrobial humoral response with production of antibacterial peptides, autophagosome assembly, response to antibiotic and prostaglandin biosynthesis. Decreased functions that were not specific to any group and characteristic to all treatments were neutrophil activation, cytokine production and response to IL-4. More decreased adaptive immune system and lymphoid cells-related functional terms were observed in lesional lung tissue from group 1: positive regulation of B cell proliferation, adaptive immune response, positive regulation of interleukin-17 production. Also in group 4 (FPI day 5–10)–T cell co-stimulation, and in group 6 (FPI and Ibuprofen day3-10)–antigen presenting functions were reduced. Type one interferon production was reduced in LL of groups 5 and 6. In group 2 (Ibuprofen day 5–10) reduction of interferon-gamma production and medium-chain fatty acid metabolism was observed.

Supplementary KEGG pathway enrichment analysis revealed 17 significantly enriched pathways (Fig 4 and Table 4). Most of the pathways were related to immune functions and response to pathogens. Non-specific pathways enriched in all treatment groups were: *Staphylococcus aureus* infection, complement and coagulation cascades and Fc gamma R-mediated phagocytosis. All of these functions were reduced in LL compared to the LN. The greatest number of pathways were enriched in group 6, including a series of connected terms, such as antigen processing and presentation, graft-versus-host disease, those pathways associated with some non-viral diseases (inflammatory bowel disease, leishmaniasis, toxoplasmosis). Also, 2 pathways that mapped nearby were Pertussis and Toll-like receptor signaling. Another 2 KEGG pathways connected to each other and characteristic to group 6 were B cell receptor signaling pathway and Fc epsilon RI signaling pathway. Expression of genes encoding most of the molecules belonging to these pathways was lower in LL. In group 6 only one pathway demonstrated increased function; it was autophagy. Increased chemokine signaling pathway was observed in LL of groups 1 and 6.

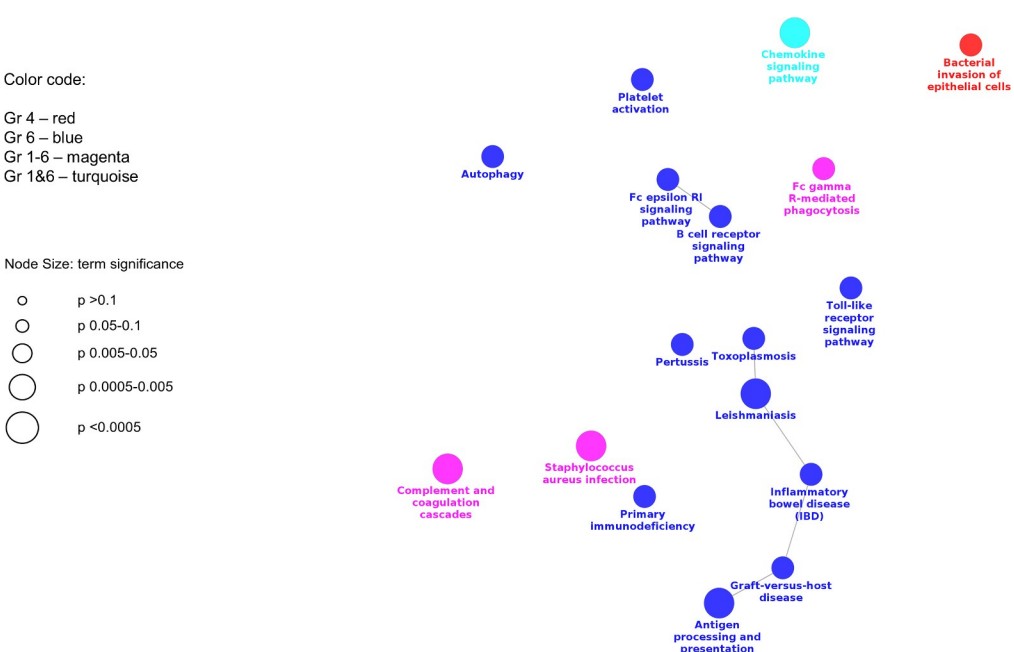

**Fig 4. Enriched KEGG pathways related to the immune functions in lung lesion (LL) vs lung without lesions (LN) is shown.** KEGG pathway enrichment analysis of differentially expressed genes (p<0.05) in each treatment group was conducted and results were visualized as a map of predicted connections among significantly enriched pathways.

**Table 4. Enriched KEGG pathways related to the immune functions in lung lesions.**

| Treatment Group | Increased function | Decreased function |
|---|---|---|
| 1 –Ibuprofen, day 3–10 | No pathways | No pathways |
| 2 –Ibuprofen, day 5–10 | No pathways | No pathways |
| 3 –Placebo | No pathways | No pathways |
| 4 –FPI, day 5–10 | No pathways | Bacterial invasion of epithelial cells |
| 5 –FPI + Ibuprofen, day 5–10 | No pathways | No pathways |
| 6 –FPI + Ibuprofen, day 3–10 | Autophagy | Antigen processing and presentation |
| | | Leishmaniasis |
| | | Inflammatory bowel disease (IBD) |
| | | B cell receptor signaling pathway |
| | | Graft-versus-host disease |
| | | Fc epsilon RI signaling pathway |
| | | Pertussis |
| | | Primary immunodeficiency |
| | | Toll-like receptor signaling pathway |
| | | Platelet activation |
| | | Toxoplasmosis |
| Nonspecific pathways (Enriched in groups 1 and 6) | Chemokine signaling pathway | No pathways |
| Nonspecific pathways (enriched in all groups) | No pathways | Staphylococcus aureus infection |
| | | Complement and coagulation cascades |
| | | Fc gamma R-mediated phagocytosis |

## Differential gene expression in lung lesions among treatment groups

When lung lesion (LL) samples were compared across treatment groups in all combinations, only one gene was differentially expressed in the group 3 (placebo) vs group 2 (Ibuprofen day 5–10) comparison (ENSBTAG00000025952, no gene name (odorant-binding protein [Source: NCBI gene;Acc:785083]), two genes were differentially expressed in the comparison of group 5 (FPI and Ibuprofen day 5–10) vs group 2 (ENSBTAG00000025952 and CCDC113) and 94 genes were differentially expressed in lung lesion samples in the comparison between group 6 (FPI and Ibuprofen day 3–10) and group 2. Enrichment analysis in this set of genes revealed 21 enriched GO terms with p<0.05 (S3 File, analysis was performed with GO term redundancy reduction by merging GO terms with 50% of overlap). Among them, 12 GO terms deserve special attention, because they were mapped into a network of functionally associated terms with multiple shared genes (Fig 5). All these terms are related to cilia formation, assembly, structure, motility and regulation. Interestingly, in lung without lesions (LN), among all treatment groups comparison combinations only comparison of Gr 6 vs Gr 2 showed 12 differentially expressed genes, but these were different from the set of genes in lung lesions (LL). GO enrichment functional analysis revealed a group of 5 GO terms (with adjusted p<0.05) combined into a strong network with shared genes and all the terms were related to chromosome structure and functioning (S2 Fig). After the fusion of redundant terms by ClueGo they formed 2 terms: "metaphase plate congression" and "condensed chromosome kinetochore". Another group of involved genes, separate from this group term, was "basement membrane".

## GO enrichment analysis in lung tissue with- and without a lesion in all treatment groups in comparison with the placebo group

A major goal of this research was to demonstrate differences in the functional analysis between lung tissue with lesions (LL) and lung tissue without lesions (LN) in the context of the

Node Color: significance

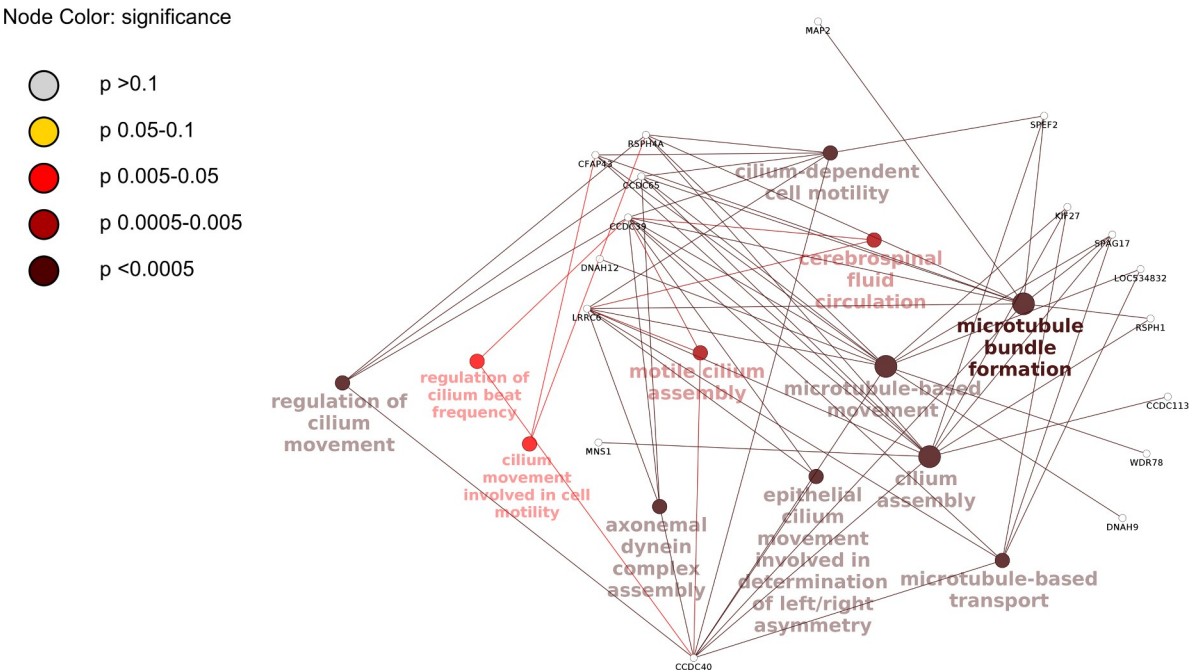

**Fig 5. Cluster of functionally connected GO terms in lung lesions (LL) and their associated genes in the comparison of treatment group 6 to group 2.** Differential gene expression analysis in LL across treatment groups revealed DE genes only in treatments group 6 versus 2. Gene ontology enrichment analysis of this gene list (p<0.05) was conducted and results were visualized as a map of predicted connections among significantly enriched GO terms and related genes.

comparison of each treatment group with the placebo group. GO terms were sorted based on type of tissue sample, treatment group and functional clusters. We combined GO terms to the following functional clusters relevant to this study: "Innate immune functions", "Adaptive immune functions", "Prostanoid synthesis and fatty acid metabolism", "Oxidative stress", "Damage and starvation", "Vasculogenesis and regeneration" (Table 5). Only GO terms with p<0.05 were used for this selection. Interestingly, treatment group 6 had no difference between LL and LN in such innate system functions as inflammatory response, positive regulation of neutrophil chemotaxis, chemokine-mediated signaling pathway, response to cytokines and negative regulation of viral genome replication. As for the other groups in the innate immune functions cluster, Group 1 had enriched terms for LL in activation of innate immune response, MAPK kinase, macrophage differentiation and several GO terms related to neutrophil activation, cell-cell adhesion and motility. In contrast, in group 2 innate immune function-related enrichment was observed mostly in LN, including: LPS-mediated signaling pathway, inflammatory response, defense response to virus, antigen processing and presentation by MHC class I. In group 4 LL had an enriched inflammatory response, neutrophil chemotaxis, chemokine signaling pathway, integrin adhesion, mast cell degranulation, cell migration and cytokine production. The last two terms were also enriched in LN in group 4. Other LN associated terms in this group were viral process, positive regulation of TNF alpha production and TLR-4 signaling pathway. Among innate immune functions specific to LN in treatment group 5, positive regulation of IL-8 production and positive regulation of I-kappa B kinase/NF-kappaB signaling were observed. Functions enriched in both types of lung tissue of group 5 were regulation of immune response and induction of positive chemotaxis.

Adaptive immune functions were mostly enriched in the comparisons of lung tissue without lesion: in group 4 –T cell activation, groups 5 and 6 –T cell co-stimulation, group 2 –Th2

**Table 5. GO enrichment analysis in lung tissue with- and without lesion in all treatment groups in comparison with the placebo group.**

| GO ID | GO Terms | Compared to Placebo | |
|---|---|---|---|
| | | lesion (Gr#) | no lesion (Gr#) |
| | *Innate immune functions* | | |
| GO:0002218 | activation of innate immune response | 1, 2 | |
| GO:0032874 | positive regulation of stress-activated MAPK cascade | 1, 5 | |
| GO:0010744 | positive regulation of macrophage derived foam cell differentiation | 1, 4 | |
| GO:0031663 | lipopolysaccharide-mediated signaling pathway | | 2, 6 |
| GO:0042590 | antigen processing and presentation of exogenous peptide antigen via MHC class I | | 2, 5 |
| GO:0051607 | defense response to virus | | 2, 4 |
| GO:0006954 | inflammatory response | 4, 6 | 2, 6 |
| GO:0090023 | positive regulation of neutrophil chemotaxis | 4, 6 | 5, 6 |
| GO:0070098 | chemokine-mediated signaling pathway | 4, 6 | 6 |
| GO:0042119 | neutrophil activation | 1 | |
| GO:0007159 | leukocyte cell-cell adhesion | 1 | |
| GO:0030836 | positive regulation of actin filament depolymerization | 1 | |
| GO:0030043 | actin filament fragmentation | 1 | |
| GO:0032088 | negative regulation of NF-kappaB transcription factor activity | | 2 |
| GO:0033026 | negative regulation of mast cell apoptotic process | | 2 |
| GO:0042742 | defense response to bacterium | | 2 |
| GO:0002755 | MyD88-dependent toll-like receptor signaling pathway | | 2 |
| GO:0045351 | type I interferon biosynthetic process | | 2 |
| GO:0033630 | positive regulation of cell adhesion mediated by integrin | 4 | |
| GO:0043303 | mast cell degranulation | 4 | |
| GO:0016032 | viral process | | 4 |
| GO:0032760 | positive regulation of tumor necrosis factor production | | 4 |
| GO:0034142 | toll-like receptor 4 signaling pathway | | 4 |
| GO:0034145 | positive regulation of toll-like receptor 4 signaling pathway | | 4 |
| GO:0045080 | positive regulation of chemokine biosynthetic process | 2 | 2 |
| GO:0001816 | cytokine production | 4 | 4 |
| GO:0016477 | cell migration | 4 | 4 |
| GO:0032757 | positive regulation of interleukin-8 production | | 5 |
| GO:0043123 | positive regulation of I-kappaB kinase/NF-kappaB signaling | | 5 |
| GO:0050930 | induction of positive chemotaxis | 5 | 5 |
| GO:0050776 | regulation of immune response | 5 | 5 |
| GO:0050930 | induction of positive chemotaxis | 6 | |
| GO:0050776 | regulation of immune response | 6 | |
| GO:0050766 | positive regulation of phagocytosis | | 6 |
| GO:0071346 | cellular response to interferon-gamma | | 6 |
| GO:0071347 | cellular response to interleukin-1 | | 6 |
| GO:0006958 | complement activation, classical pathway | | 6 |
| GO:0007250 | activation of NF-kappaB-inducing kinase activity | | 6 |
| GO:0030593 | neutrophil chemotaxis | | 6 |
| GO:0045071 | negative regulation of viral genome replication | 6 | 6 |
| GO:0034097 | response to cytokine | 6 | 6 |
| | *Adaptive immune functions* | | |
| GO:0042110 | T cell activation | 2, 6 | 4 |
| GO:0031295 | T cell costimulation | | 5, 6 |
| GO:2000553 | positive regulation of T-helper 2 cell cytokine production | | 2 |

*(Continued)*

**Table 5.** (Continued)

| GO ID | GO Terms | Compared to Placebo | |
|---|---|---|---|
| | | lesion (Gr#) | no lesion (Gr#) |
| GO:2000401 | regulation of lymphocyte migration | | 5 |
| | *Prostanoid synthesis and fatty acid metabolism* | | |
| GO:0042761 | very long-chain fatty acid biosynthetic process | 1, 4 | |
| GO:0006631 | fatty acid metabolic process | | 1, 4, 6 |
| GO:0001516 | prostaglandin biosynthetic process | 4 | 1 |
| GO:0070328 | triglyceride homeostasis | 1 | |
| GO:0006631 | fatty acid metabolic process | | 1 |
| GO:0030497 | fatty acid elongation | 4 | |
| | *Oxidative stress* | | |
| GO:0042744 | hydrogen peroxide catabolic process | 2, 4 | |
| GO:0098869 | cellular oxidant detoxification | | 2, 6 |
| GO:0006979 | response to oxidative stress | 4 | 6 |
| GO:0042542 | response to hydrogen peroxide | 4, 5, 6 | 4 |
| GO:0045454 | cell redox homeostasis | 1 | |
| GO:0034614 | cellular response to reactive oxygen species | 1 | |
| GO:0032981 | mitochondrial respiratory chain complex I assembly | 2 | |
| GO:0034599 | cellular response to oxidative stress | 2 | |
| GO:0035694 | mitochondrial protein catabolic process | 2 | |
| GO:0019430 | removal of superoxide radicals | | 2 |
| GO:2000377 | regulation of reactive oxygen species metabolic process | | 5 |
| | *Damage and starvation* | | |
| GO:1902902 | negative regulation of autophagosome assembly | 4 | |
| GO:0006508 | Proteolysis | 4 | 4 |
| GO:0043161 | proteasome-mediated ubiquitin-dependent protein catabolic process | | 5 |
| GO:0097352 | autophagosome maturation | 2 | |
| | *Vasculogenesis and regeneration* | | |
| GO:0090051 | negative regulation of cell migration involved in sprouting angiogenesis | 1 | |
| GO:0030148 | sphingolipid biosynthetic process | 1 | |
| GO:0044344 | cellular response to fibroblast growth factor stimulus | 1 | |
| GO:0090156 | cellular sphingolipid homeostasis | | 2 |
| GO:1901888 | regulation of cell junction assembly | 4 | |
| GO:0010575 | positive regulation of vascular endothelial growth factor production | 5 | |
| GO:0001570 | Vasculogenesis | 6 | |
| GO:0060055 | angiogenesis involved in wound healing | 6 | |
| GO:2001214 | positive regulation of vasculogenesis | | 6 |
| GO:0016525 | negative regulation of angiogenesis | | 6 |

cytokine production and group 5 –regulation of lymphocyte migration. The only function that was enriched in lung tissue lesions (in groups 2 and 6) was T cell activation.

The prostaglandin biosynthetic process was enriched in lung lesions of treatment group 4 and in lung tissue without lesions in group 1. Enrichment of other terms connected to fatty acid metabolism was also observed, predominantly in these 2 groups in both types of lung tissue samples.

Oxidative stress-related terms were enriched mainly in samples from lung tissue with lesion in all treatment groups in the comparison to the placebo. The only terms that were enriched in lung tissue without lesion were: cellular oxidant detoxification (groups 2 and 6), response to

oxidative stress (group 6), response to hydrogen peroxide (group 4), removal of superoxide radicals (group 2) and regulation of reactive oxygen species metabolic process (group 5).

Autophagy and catabolic processes were predominantly enriched in lung tissue with lesion of groups 4 and 2, as well as vasculogenesis and regeneration terms, especially in groups 1, 4, 5 and 6.

## KEGG pathway analysis in lung tissue with- and without lesion in all treatment groups in comparison with the placebo group

KEGG pathway enrichment analysis of the whole transcriptome was performed in the context of comparison of every treatment group with the placebo and how this result is different in lung tissue with lesion and lung tissue without lesion. KEGG pathways with p<0.05 and relevant to the study were selected and grouped (similar to GO terms) into several functional categories: "Innate immune functions", "Adaptive immune functions", "Complex immune pathway or response to pathogens", "Prostanoid synthesis and fatty acid metabolism", "Oxidative stress", "Cell adhesion".

For innate immune functions (Table 6) differences between treatment and placebo dominate in lung tissue without lesion. RIG-I-like receptor signaling pathway enrichment was observed in this sample type for all groups; NF-kappa B signaling pathway was also enriched in all groups except group 1. Groups 1 and 6, which were treated with ibuprofen for the longest time, demonstrated enrichment in natural killer cell mediated cytotoxicity and Complement and coagulation cascades. Complement and coagulation cascades pathway was also enriched in lung tissue with lesion of all treatment groups, except group 4 (treated with FPI day 5–10). Antigen processing and presentation pathway enrichment was unique for only lung tissue with lesion in groups 4, 5 and 6 (all FPI treated groups). In the cluster of adaptive immune functions most of the pathways were enriched in lung tissue without lesion for group 6. T cell receptor signaling pathway was enriched only in this tissue type in groups 1 and 6, which had the longest ibuprofen treatment. IgA production was enriched in lung tissue with lesion for groups 4 and 6 (FPI treated); and in group 6 IgA production was also enriched in lung tissue without lesion. IL-17 signaling and Th17 differentiation pathways were mainly observed in groups 5 and 6 (combined treatment). Th1 and Th2 cell differentiation enrichment was uniquely attributed to group 6 in both types of lung tissue.

Interestingly, most of the complex immunopathology pathways or response to pathogen pathways were enriched in group 6 and often in both LL and LN, for example: asthma, allograft rejection, graft-versus-host disease. Among all pathways, *Staphylococcus aureus* infection was the most common enriched pathway for most groups (groups 1, 2, 6 –in LL and 1 and 6 –in LN). Measles pathway enrichment was also common among most of the treatment groups, but only in LN.

In terms of prostanoid synthesis the most important pathway that was enriched in multiple treatment groups is arachdonic acid metabolism. It was enriched in lung tissue with lesion samples from treatment groups 1, 4, and 6, as well as in lung tissue without lesion in groups 2 and 6.

Peroxisome pathway was enriched in lung tissue without lesion in group 4. This was the only enriched pathway directly related to the oxidative status.

Cell adhesion and tight junctions were affected in lung tissue with lesion from groups 1, 2 and 4; and also gap junction in group 2. Most of the genes associated with these GO terms showed reduced expression in these treatment groups.

## Results of microbiology culture and viral load

There was no consistent pattern of bacterial infection in any animals in all treatment groups. There were rare to moderate numbers of bacteria cultured in some animals. Most of the

**Table 6. KEGG pathway enrichment analysis in lung tissue with- and without lesion in all treatment groups in comparison with the placebo group.**

| KEGG ID | KEGG pathway | Compared to Placebo | |
|---|---|---|---|
| | | lesion (Gr) | no lesion (Gr) |
| | *Innate immune functions* | | |
| KEGG:04611 | Platelet activation | 1 | 5 |
| KEGG:04015 | Rap1 signaling pathway | 1, 4 | |
| KEGG:04621 | NOD-like receptor signaling pathway | | 1, 2 |
| KEGG:04622 | RIG-I-like receptor signaling pathway | | 1, 2, 4, 5, 6 |
| KEGG:04650 | Natural killer cell mediated cytotoxicity | | 1, 6 |
| KEGG:04610 | Complement and coagulation cascades | 1, 2, 5, 6 | 1, 6 |
| KEGG:04670 | Leukocyte transendothelial migration | 1, 2, 4 | 1 |
| KEGG:04062 | Chemokine signaling pathway | 1, 4, 5 | 1, 2, 6 |
| KEGG:04060 | Cytokine-cytokine receptor interaction | 4, 6 | 2, 6 |
| KEGG:04064 | NF-kappa B signaling pathway | | 2, 4, 5, 6 |
| KEGG:04620 | Toll-like receptor signaling pathway | | 2, 4 |
| KEGG:04623 | Cytosolic DNA-sensing pathway | | 2, 4 |
| KEGG:04612 | Antigen processing and presentation | 4, 5, 6 | |
| KEGG:04145 | Phagosome | | 4, 5 |
| KEGG:04750 | Inflammatory mediator regulation of TRP channels | 4 | 6 |
| KEGG:04630 | Jak-STAT signaling pathway | 1 | |
| KEGG:04666 | Fc gamma R-mediated phagocytosis | 1 | |
| KEGG:04668 | TNF signaling pathway | | 2 |
| | *Adaptive immune functions* | | |
| KEGG:04662 | B cell receptor signaling pathway | 1, 5 | |
| KEGG:04660 | T cell receptor signaling pathway | | 1, 6 |
| KEGG:04657 | IL-17 signaling pathway | | 2, 5, 6 |
| KEGG:04672 | Intestinal immune network for IgA production | 4, 6 | 6 |
| KEGG:04659 | Th17 cell differentiation | 5, 6 | 6 |
| KEGG:04658 | Th1 and Th2 cell differentiation | 6 | 6 |
| | *Complex immune pathway or response to pathogens* | | |
| KEGG:05162 | Measles | | 2, 4, 5, 6 |
| KEGG:05164 | Influenza A | | 2, 6 |
| KEGG:05168 | Herpes simplex infection | | 5, 6 |
| KEGG:05323 | Rheumatoid arthritis | 1, 5 | 6 |
| KEGG:05150 | Staphylococcus aureus infection | 1, 2, 6 | 1, 6 |
| KEGG:05133 | Pertussis | 6 | 2, 4 |
| KEGG:05134 | Legionellosis | | 2, 4 |
| KEGG:05321 | Inflammatory bowel disease (IBD) | | 6 |
| KEGG:05167 | Kaposi's sarcoma-associated herpesvirus infection | 1 | |
| KEGG:05416 | Viral myocarditis | 6 | |
| KEGG:05161 | Hepatitis B | | 6 |
| KEGG:05310 | Asthma | 6 | 6 |
| KEGG:05330 | Allograft rejection | 6 | 6 |
| KEGG:05332 | Graft-versus-host disease | 6 | 6 |
| | *Prostanoid synthesis and fatty acid metabolism* | | |
| KEGG:01040 | Biosynthesis of unsaturated fatty acids | 1 | |
| KEGG:00590 | Arachidonic acid metabolism | 1, 4, 6 | 2, 6 |
| KEGG:00062 | Fatty acid elongation | 1, 4 | |
| KEGG:00071 | Fatty acid degradation | | 4 |

*(Continued)*

**Table 6.** (Continued)

| KEGG ID | KEGG pathway | Compared to Placebo | |
|---|---|---|---|
| | | lesion (Gr) | no lesion (Gr) |
| KEGG:00592 | alpha-Linolenic acid metabolism | | 4 |
| | *Oxidative stress* | | |
| KEGG:04146 | Peroxisome | | 4 |
| | *Cell adhesion* | | |
| KEGG:04530 | Tight junction | 1, 2, 4 | |
| KEGG:04540 | Gap junction | 2 | |
| KEGG:04514 | Cell adhesion molecules (CAMs) | 6 | 6 |

positive cultures were from the bronchial swabs. The following isolates were registered (with number of positive samples): *Streptococcus* spp.– 4; *Arcanobacterium pyogenes*– 2; *Streptococcus gallolyticus*– 1; *Klebsiella oxytoca*– 1; *Pasteurella multocida*– 1; *Mannheimnia hemolytica*– 1; *Acinetobacter wolfi*– 1; *Staphylococcus xylosus*– 1; *Biebersteinia trehalosi*– 1; *Mannheimnia granulomatis*– 1; *Streptocossus gallolyticus*– 2; *Comamons kersterisi*– 1; *Corynebacteria spp.* - 1; *Truperella pyogenes*– 1; *Kocuria rhizophila*– 1; rare mixed flora– 9.

There was no apparent bacterial infection in the lungs of any of the calves in the study, i.e. no purulent exudate. The isolated bacteria were diverse and infrequent. *Mannheimnia hemolytica*, *Pasteurella multocida*, and *Biebersteinia trehalose* would be the most common respiratory bacteria to be found in calves; while some of the other bacteria were likely contaminants.

As previously described [19] viral shedding from nasal swab qRT-PCR measurements showed the highest levels of virus particles shed for group 1 followed by group 2 and placebo with peak shedding on days 5–6, and a return to baseline by day 10 when necropsy was performed and tissues were sampled for pathology and RNA sequence analysis. Group 6 had significantly less viral shedding than any other group.

## Results of gross lung consolidation evaluated at necropsy

As stated previously lung pathology is described elsewhere for this study. To briefly summarize here: the mean percent lung consolidation for each treatment group, as determined by a board certified veterinary pathologist (FRC), was: group 1–22.5%; group 2–33.1%; group 3–31.5%; group 4–24.5%; group 5–25% and group 6–9.3%. Histological observations when scored as absent (0), mild (1), moderate (2), or severe (3) for nine different lesions in six compartments (pleura, alveolus, septum, interstitium, bronchiole, and bronchus) demonstrated significant differences for both treatment drug(s) and start times that were significant, including highly significant lower scores for group 6, treated with both FPI and ibuprofen from day 3 when compared to other groups. Specific histological findings that were scored for each compartment varied by compartment and included: monocytic infiltrates, neutrophilic infiltrates, syncytial cells, epithelial transmigration, peribronchiolar lymphoid and monocytic infiltrates, epithelial cell necrosis, deciliation and lymphoid nodules, as well as other compartment specific observations.

## Discussion

The goal of this study was to evaluate the effect of different treatment approaches on functional changes and immune status of the lung tissue in calves infected with BRSV. Previously, to the best of our knowledge results of differential gene expression between healthy lungs and BRSV infected lungs were reported in only one study [20]. Behura et al was carried out using the

same BRSV isolate (CA-1) and a similar mode of aerosol infection and allows some comparability of results, Behura et al examined differential gene expression in lung tissue (both lesional and non-lesional) in BRSV infected and uninfected untreated bovines. There are other differences.: Behura et al used 6–8 month old Angus-sired crossbred steers and performed necropsy at the peak of clinical illness (Day 6–7). This contrasts with our use of 5–6 week old pre-ruminant Holsteins and performance of necropsy after peak illness (Day 10), We compared the effect of two therapeutic drugs given at several time intervals and to a placebo. The similar mode of infection and virus isolate used as inoculum make it possible to compare the results of these studies, although breed and age differences are factors that may explain some differences in results.

In our study the greatest number [60] of significantly differentially expressed genes between lung tissue with lesion and lung tissue without lesion occurred in the placebo group. This suggested that all treatment combinations reduced differential gene expression between these two tissue types. The placebo group (Group 3) was unique in significant expression of pathways related to stress-associated regulation of nucleic acids. One of the genes responsible for this function is DNAJB1. Its expression was significantly increased in lung tissue with lesion. It codes for DnaJ heat shock protein family (Hsp40) member B1. This protein is upregulated under stress factors such as high temperature and some chemicals, such as arsenite and azetidine carboxylic acid [28]. HSP40 is able to translocate and accumulate in nuclei of stressed cells [29]. Functionally, HSP40 is a co-chaperon of HSP70 which has antiapoptotic function and can protect hematopoietic cells from cytotoxicity induced by IFN-γ and TNF-α [30]. It is possible that this protein is essential for BRSV replication. It was previously demonstrated that HSP40/DNAJB1 is important for efficient influenza A virus replication by assisting nuclear trafficking of viral nucleoprotein [31]. We are unaware of experimental data on whether these chaperons also participate in replication of BRSV.

Reduction of stress in infected lungs of treated calves appears to be an effect of all treatments. Another stress-related gene that was differentially expressed in lung tissue with lesion is TMBIM6, which encodes BAX inhibitor 1. It is an anti-apoptotic protein protecting cells from endoplasmic reticulum associated stress which is characterized by accumulation of unfolded proteins and reactive oxygen species and can be caused by the infection and other stress-factors. BAX inhibitor 1 regulates reactive oxygen species (ROS) in endoplasmic reticulum [32]. There is a report suggesting that BAX inhibitor 1 serves as an antiviral agent in influenza virus infected cells through the suppression of reactive oxygen species mediated cell death and upregulation of heme oxygenase-1 expression [33]. Interestingly, removal of reactive oxygen species by a scavenging agent reduced replication of the influenza virus.[33]. In our study expression of the BAX inhibitor 1 was lower in lung tissue with lesion than in lung tissue without lesion suggesting that the lower level of its expression may have contributed to higher oxidative stress and led to the development of visible pathological changes. However, the cause-and-effect relationship in this process is not clear. It is possible that BRSV infection caused an undefined mechanism of suppression of the BAX inhibitor 1 or there may have been other mechanisms that caused oxidative stress, tissue damage and visible lesions. Both of these genes were differentially expressed, albeit statistically significantly only in the placebo group. This suggests that all treatment options reduce stress even in lesions of the lung of infected animals. These observations are consistent with Behuda et al who found that in tissues compared between BRSV infected steers and control steers differentially expressed genes were involved in oxidative phosphorylation, mitochondrial dysfunction, and hypoxia [20].

Results of the functional analysis of genes that were differentially expressed exclusively in treatment group 5 (ibuprofen and FPI, day 5–10), showed only one statistically significant GO term: "positive regulation of potassium ion transport". This function was apparently lower in

lung tissue with lesion, because expression of both genes related to this function (FHL1, WNK1), was lower. FHL1 is a "four-and-a-half LIM domain protein 1". It has been demonstrated that FHL1 is a major factor for replication of some viruses. This was suggested by the experimental results where FHL1 deficient cells are resistant to chikungunya virus [34]. One of the FHL1 isoforms is expressed in fibroblasts and it is likely that lung fibroblasts were a major source of FHL1 mRNA in our samples. The mechanism of how this protein works for viral replication is not fully known, besides its interaction with nonstructural nsP3 protein of chikungunya virus; and it is not known whether FHL1 has a direct effect on BRSV replication and how the combination of FPI and ibuprofen may have suppressed FHL1 expression.

Among 6 genes that were differentially expressed and unique for only group 6, in which animals were treated with both ibuprofen and FPI, starting on day 3 after the infection, we have found two genes directly responsible for immune functions. One of them was CD180. It is a radioprotective 105 kDa (RP105) protein, which is an unconventional member of the TLR family expressed on the cell surface. As reviewed [35], unlike typical TLR, RP105 does not have an intracellular Toll-IL-1R signaling domain and is expressed on macrophages, dendritic cells and B lymphocytes. However, this protein has the opposite role in B cells versus myeloid cells. In B cells RP105 promotes activation and also works as an anti-apoptotic factor, preventing radiation- and steroid-induced apoptosis. In contrast, in myeloid cells RP105 is associated with suppression of an inflammatory cytokine response to LPS. It is also known that certain molecular patterns, like mycobacterial lipoprotein, activate macrophages through RP105. These dual roles of RP105 may be explained by different coreceptors of RP105 in different cells, such as CD19 that is expressed only on B cells. In myeloid cells only TLR2 and TLR4 are coreceptors of RP150. As for the role of this molecule in viral infections, up-regulated expression of RP150 in B cells of patients with influenza was previously reported [36]. In our study expression of RP150 was lower in lesional than non-lesional lung. The role of BRSV in the expression of this protein and the potential effect of the therapeutics remain to be discovered. One possible explanation pertains to the relatively late stage of infection (day 10) when samples were taken and thus RP105 expression may be a result of an anti-inflammatory (pro-resolving) environment within lesion areas.

Another gene that was differentially expressed only in the treatment group 6 was CMTM7, which is a "CKLF-like MARVEL transmembrane domain containing 7". It belongs to chemokine-like factors super family (CKLFSF). It has been demonstrated that lack of CMTM7 causes reduction in innate-like B cell population B-1a leading to a deficiency of natural IgM and IL-10 [37]. Considering the fact that CMTM7 expression in lung tissue with lesion was lower, we can only speculate that its expression was reduced in lesions because the presence of B-1a cells was no longer required in resolving lung tissue. In contrast it may be upregulated in lung tissue without lesion to attract and support innate-like B cells.

Another gene that deserves special attention is "purine nucleoside phosphorylase" or PNP. Changes in the expression of this gene may have potential clinical implications. Expression of this gene was lower in lung tissue with lesion. It may have had a protective antiviral effect to reduce viral replication by slowing down purine metabolism, or it may have had a negative effect on kinetics of antiviral drugs such as ribavirin. This antiviral has an effect on many RNA viruses including RSV, by inhibition of RNA synthesis [38]. The antiviral used in this study has a different mechanism of action than ribavirin but some mechanisms by which the host responds to this antiviral may be in common. PNP is a key enzyme required for bioavailability of ribavirin [39]. Lower PNP levels in lung tissue with lesion may reduce phosphorylation of ribavirin and impair the antiviral effect of this drug. However, reduced expression of PNP is most likely a part of the cellular intrinsic virus- or stress-induced protection mechanism. Additionally, expression of ADA (adenosine deaminase), which is also responsible for the purine

metabolism, was significantly lower in lung tissue with lesion in all treatment groups. It has been demonstrated that adenosine deaminase enzyme acting on RNA 1 (ADAR-1) is responsible for interactions with some RNA viruses and supports their replication [40], but there is no data suggesting any effect of ADA on the BRSV replication.

Comparative GO enrichment analysis of whole lists of differentially expressed genes did not reveal any treatment group-specific terms or pathways except for the regulation of transcription in response to stress that was discussed above. Generally, for all treatment groups GO functions corresponding to the immune system were dominant. That includes B cell activation and differentiation, regulation of B cells and antigen receptor signaling. We demonstrated that virus shedding on day 10 was very low [19], but the B cell response in lung tissue was still active with the greatest differences being between lung tissue with lesion versus lung tissue without lesion. ADA and also MFNG and/or CMTM7 genes were involved in these functions. ADA and CMTM7 were discussed before and our result suggested that their differential expression was the result of suppression of the adaptive B cell response and, possibly, other immune functions. MFNG gene encodes O-fucosylpeptide 3-beta-N-acetylglucosaminyltransferase. This protein is part of notch signaling pathway and it has been demonstrated that, besides B cells [41], it plays some role in Th cell differentiation and its overexpression may cause enhanced Th1 differentiation [42]. Expression of this gene was also lower in lung tissue with lesion and all these results together suggest that adaptive immune functions in this tissue type were either suppressed or the presence of lymphoid cells was lower than in lung tissue without lesion.

The same GO enrichment analysis also demonstrated differences in the innate immune response in lung tissue with lesion and lung tissue without lesion. One of these functions was granulocyte activation. Genes ANXA3 and/or FCGR3A were differentially expressed and levels of their expression were also significantly lower in lung tissue with lesion. This finding is also in concordance with significantly lower expression of complement components in lesions. This is another fact which suggests that immune mechanisms in lung lesions are impaired due to the stress or simply because immune cells are no longer attracted to these sites. KEGG pathway enrichment analysis of this gene set confirmed reduced immune function and even "Primary immunodeficiency" pathway (a collection of genes that are affected in cases of primary immunodeficiency and are therefore grouped under this heading) was enriched together with the "complement and coagulation cascades pathway". Notably, genes that belong to GO terms and pathways, related to the vascular development also showed relatively lower expression, suggesting that despite that it was the $10^{th}$ day after virus inoculation, restoration of the tissue in these parts of the lung was not as active as it was in lung tissue sites without lesion. It also can be explained by reduced function or presence of myeloid cells, such as macrophages in the lesion sites, and these cells play a central role in regulation of vascular remodeling and restoration of the lung tissue [43].The overall scheme identified by analysis of differential gene expression in lung tissue with lesion in comparison with lung tissue without lesion may be characterized as a stress or oxidative stress with increased chaperon/heat shock proteins and depression of other functions, including all branches of the immune response. These stress related markers were reduced by all combinations of treatment. This result was based not only on functional analysis of differentially expressed genes but also on conducting GO enrichment analysis using data of the entire transcriptome. Our analysis is not limited to lists of only differentially expressed genes and uses data of all expressed genes to allow revelation of a broader picture and determination of general trends in the functional differences among treatment groups. This analysis confirmed some of the conclusions based on the differential gene expression, for example: depressed activation and differentiation of B cells, T cells, cytokine production and adaptive immune response generally in lesions. However, chemotaxis of immune

cells, especially innate immune cells, was increased in lesion sites of most of the treatment groups. Differential gene expression suggested that there is probably a shortage or depressed functional activity of innate immune cells in lesions and increased chemotaxis is probably a local compensatory effect in response to the shortage.

Results of this analysis in the placebo group were also in agreement with the differential gene expression analysis and confirmed that the most characteristic unique difference between lung tissue with lesion and lung tissue without lesion was oxidative stress if no treatment was applied. Depressed regulation of the protein processing in this tissue may indicate greater stress and possible starvation or more significant functional suppression in cells in lesions without treatment.

Calves treated with ibuprofen alone showed increased viral shedding in a previous study from this lab as well as in the present study [15, 19]. Thus it is an interesting finding that increased function of "negative regulation of viral genome replication" in groups 1 and 2, as well as "Defense response to virus" enriched in group 2 was found. This may be connected with our results of virus shedding measurements in this and in the previous experiment because these groups were treated only with ibuprofen and maximum viral loads were detected [19]. The viral shedding was significantly higher than in animals in the placebo group as reported in this study [19] and in our previous work [15]. Apparently, the anti-inflammatory or antipyretic function of ibuprofen was favorable for viral replication. As a result, increased antiviral function is probably a response to increased virus replication, caused by ibuprofen.

One of the genes that was responsible for the enrichment of the "Defense response to virus" function was RSAD2 (Viperin) which was 2.4 logFC higher in lung tissue with lesion. This finding is also apparent in Behura et al. It identified differential expression of the RSAD2 gene in the BRSV infected steers in that study compared to uninfected normal control steers. Its finding was also in lung with lesion [20]. RSAD2 expression is significantly upregulated by human RSV infection *in vitro* and *in vivo* and it had an inhibitory effect on viral replication [44]. It was also demonstrated that viperin production in bovine epithelial cells was increased with bovine RSV infection which caused reduction in virus replication [45]. RSAD2 belongs to the group of interferon-stimulated antiviral genes and depends on type I interferons and active involvement of the JAK-STAT signaling pathway. Antiviral mechanisms of viperin are characterized by perturbed lipid rafts and promoted TLR7 and TLR9 signaling as reviewed [46]. Considering direct connection of RSAD2 with Type I interferons, we can see the opposite picture in group 6, where treatment was a combination of FPI and ibuprofen and started early (on day 3 after the infection). Viral load in this group was minimal and functions of "Type I interferon biosynthetic process" in lung lesions of this group were already down-regulated on day 10 after the infection. All this suggests that lung lesions are places of initial intensive replication of the virus where more extensive damage and oxidative stress impairs not only cells of local epithelial and stromal tissue, but also cells of innate and adaptive arms of the immune system and their functions.

= RSV infection causes oxidative stress in lung tissue [47] and potently induces activation of cytoplasmic mitogen- and stress-related kinase 1 (MSK1) by increased reactive oxygen species [48]. This oxidative stress itself is likely a strong regulator of gene expression, including upregulation of the TLR3 and NF-κB-related genes [49]. Our work is consistent with the concept of oxidative stress as a potent marker of the viral infection that leads to cytokine responses.

KEGG pathway enrichment analysis using the whole transcriptome, demonstrated that most of the enriched pathways were specific to group 6 (FPI and ibuprofen day 3–10), where we can see suppression of immune functions, including response to pathogens. It is important to mention that the "Fc epsilon RI signaling" pathway was suppressed in group 6. This

mechanism is responsible for IgE-mediated activation of mast cells and is very characteristic of respiratory syncytial virus infection as reviewed [50]. Switching the immune response to Th2 and IgE is considered to be a mechanism the RSV/BRSV evasion of the innate and cellular immune mechanisms. In our study, suppression of Fc epsilon signaling reduces consequences related to activation and degranulation of mast cells and basophils. There is a direct connection between Fc epsilon signaling and "Prostaglandin biosynthesis process" pathway, which was not enriched in group 6, but it was enriched in treatment groups 1 and 4. Prostaglandin E synthase was increased in lesions within the interval of 1.5–1.8 logFC in these 2 treatment groups but this difference was not statistically significant. It is important to note that group 5 differs from group 4 (monotherapy with FPI only, but initiated on the same day). The clinical outcome of this study showed a significantly lower score for group 5 in comparison to placebo. In contrast group 4 failed to show a significantly lower clinical score [19]. In group 6 indication of the decline of inflammatory functions and response to pathogens was most prominent. This observation is not unexpected because depression of viral replication beginning on day 3 by the FPI antiviral diminished the stimulus for inflammatory functions.

## Observations made in differential gene expression between lung lesion tissue of different treatment groups

There was no difference among all combinations of the treatment group comparisons except for the lung lesion tissue in group 6 when compared with group 2 with 94 differentially expressed genes. Functional GO enrichment analysis showed that there is a distinct group of genes responsible for multiple closely connected biological processes responsible for microtubule formation and microtubule-based cilium movement. The most common shared genes were CCDC39 and CCDC40. These and many other genes related to these functions were significantly upregulated in lesions of group 6, in which viral shedding, clinical scores and lung pathology were lowest through the course of the disease due to the most effective combination and earliest start of the treatment. Mucociliary clearance-mediated innate immunity is an important component of antiviral protection in the lung and RSV both reduces the number of motile cilia and causes detachment of ciliated epithelial cells [51, 52]. In our study, it is likely that an increase in expression of genes responsible for cilia and their motility in the treatment group in which BRSV caused minimal damage, played an important role in lung defense. However, group 1, similar to the group 2 by type of treatment but initiated two days sooner, showed similar levels of viral load and clinical scores to group 2 and did not demonstrate such a difference in the expression of cilia-related genes. Differences in the timing of treatment initiation was likely responsible for this difference.

## Observations made in differential gene expression between non-lesional lung tissue of different treatment groups

Similarly, analysis of differential gene expression of the lung tissue without lesions across treatment groups also showed comparable differences in the group 6 versus group 2 in cytoskeleton- and microtubule-related genes. but these were primarily related to the chromosome centromeric region.

Our results are also consistent with Ampuero et al that demonstrated microtubule cytoskeleton organization consistent with mitosis occurring in RSV-infected lung epithelial cells at 96 hours after infection. [53]. This suggests that in group 6 processes of mitosis started earlier than in group 2 (which had higher viral loads as shown by increased shedding) and tissue remodeling, characterized by cell proliferation and also basement membrane related processes, were already more active in the group with the lowest viral load and disease manifestation.

Mucociliary clearance, an innate immune function, is an important mechanism for keeping the lung free of inhaled pathogens and debris. It has been demonstrated that CENPF, one of the genes that was differentially expressed, encoding a centromere protein, is involved not only in metaphase chromosome support, but also in cellular cilia formation by orientation of microtubules [54–56]. This makes it connected to mucociliary clearance and similar to what was happening in lung lesions also in group 6 when compared with group 2. Therefore, similar changes were observed in lung tissue with lesion and lung tissue without lesion, in group 6 when compared to group 2; these were characterized by cilia restoration and tissue remodeling with a more apparent difference in lung lesions.

Examination of the functional gene ontology and KEGG pathway enrichment analysis of the transcriptome in the lung tissue in each treatment group when compared to the placebo group revealed interesting data on the innate immune system. For example, there was no GO category related to the innate immune response detected in lung without lesion in group 1 (ibuprofen day 3) that could distinguish it from lung lesion of the placebo group. But there are some innate functions in lung lesions of group 1 that differentiate it from lung lesions in placebo; these include complement and coagulation cascades, leukocyte transendothelial migration, and chemokine signaling pathways–all important aspects of the innate immune system. We may surmise that early activation of innate immune defenses by treatment with ibuprofen starting on the third day after BRSV infection provided an immune advantage to treated calves as compared to their placebo treated comtemporaries in group 3. Ibuprofen inhibits the production of prostaglandins and would have decreased lipid mediators in treatment group 1 compared with placebo treated calves. These differences were apparent in lung without lesion. For group 2, in contrast, we observed even more differences between lung without lesion if compared with the placebo group, particularly in functional categories of molecular patterns activation and response to the virus. Differences between group 1 and group 2 may reflect the different scheme of ibuprofen administration: group 1 treatment started earlier in the course of the infection and, therefore, acted longer and made an earlier intervention in the prostaglandin synthesis pathways. As a consequence of the immunomodulatory effect of the ibuprofen, the differences were also reflected in some innate immune mechanisms.

It is well known that prostaglandins play an essential role in the regulation of both innate and adaptive immune systems. COX inhibition and reduction of prostaglandin E production in mucosal dendritic cells, macrophages and monocytes, not only reduces the inflammatory effect of prostaglandin E, but also disrupts other immunoregulatory mechanisms [57, 58]. All treatment groups demonstrated differences in the cell redox homeostasis and oxidative stress mostly in lung lesions, which is good evidence of the beneficial effect of the nonsteroidal anti-inflammatory agent on the redox status of cells in the site of the inflammation. It has been demonstrated that ibuprofen also acts independently of COX inhibition to disrupt signaling cascades leading to NOX2 activation, preventing oxidative damage in microglia [59].

Comparing groups 2, 4, and 5 could potentially demonstrate differences between effects of ibuprofen monotherapy, antiviral monotherapy and the dual ibuprofen and antiviral and ibuprofen therapy as these groups began their treatments on day 5 post infection. It was group 4 (FPI only) that was associated with differences from the placebo that were similar to the changes in lung lesions of group 6 and were characterized by neutrophil chemotaxis, cell migration and cytokine production. In lung tissue without lesion of group 4, innate antiviral and antibacterial responses were predominant, but in group 6 the proinflammatory response-related GO terms and pathways predominated. Both treatment modalities and initiation times varied between these groups. More tissue damage-associated functions and oxidative stress in both tissue types were observed when only FPI was administered (group 4). Notably, there was no ibuprofen involved in the treatment, but prostaglandin biosynthesis was enriched in lung

lesions when group 4 and placebo were compared. However, expression of prostaglandin E synthase, most important in this context, had no difference between group 4 and the placebo. Genes: PTGDS, PTGS1 and prostaglandin F synthase II-like had the highest differences in expression. Pathogen response pathways and complex immunological processes were identified for group 6, using the Kegg enrichment pathway. As discussed above, expression of genes, related to these pathways is decreased in group 6 and it is, most likely, a result of the lower viral load in this group. This difference in viral shedding distinguishes group 6 from placebo and thus makes it stand out in the category of response to pathogens, when compared to the placebo. In contrast group 5 received the same drugs but did not begin therapy until 2 days later on day 5. This is an important difference as our previous experience with this infection model for BRSV infection of young calves viral replication is well underway and clinical signs are usually apparent by day 5. There was some small difference in enhanced expression in immune response and chemotaxis pathways when group 5 was compared with placebo, but overall there was no significant benefit of these therapies for viral shedding or lung pathology. However, there was a significant benefit for decreased clinical signs, although less significant than for group 6 [19]. Based on these results of the transcriptome analysis we can predict that to be most effective dual therapy with FPI and ibuprofen must be initiated by day 3 or earlier if possible. Administration prior to or on day 3 is difficult both for human infants and bovine calves in natural infection cases as this timepoint is prior to the initiation of clinical signs. Potentially the use of FPI alone or with ibuprofen on day 3 (previously shown to be effective) could be initiated in cases of known or suspected exposure to BRSV/RSV [15, 17].Nonsteroidal anti-inflammatory drugs can cause ulceration of the gastrointestinal tract; and it has been shown that these chemicals are able not only to generate profound changes in physical and biochemical properties of the cell membrane [60], but also to affect tight junctions and intercellular connections through overexpression of prostasin caused by ibuprofen [61]. Cell adhesion tight junction KEGG pathways enrichment analysis showed the "Tight junction" pathway enriched when placebo was compared to groups treated with ibuprofen only and the group treated with FPI only. This enrichment of the "Tight junction" KEGG pathway in lung with lesion, most likely was not an effect of ibuprofen treatment, because this pathway was enriched in group 4, in which was not used. Cell damage and stress are likely the major factors involved here, because enrichment of this pathway was observed only in lung lesions.

## Conclusions

The highlight of our conclusions from this work is the demonstration that Ibuprofen, when used alone, negatively affected the antiviral responses which led to higher virus loads, but when used in dual therapy it enhanced the specific antiviral effect of FPI. Transcriptome analysis has shown the ability of dual therapy with FPI and Ibuprofen to reduce the effect of damaging overexpression of prostanoids and oxidative stress. We found that lung tissue with grossly apparent lesions (LL) from BRSV infected calves showed the most significant differential gene expression compared with lung tissue not showing gross lesions (LN). The most obvious difference between these two types of tissue was oxidative stress and cell damage.

Another characteristic seen in lung lesions was depressed innate and adaptive immune functions. As expected, combined treatment with FPI and Ibuprofen, when started on the third day after infection, set this treatment group far apart from the placebo, especially in regard to pathways related to the innate and adaptive immune response to the pathogen in both LL and LN. Due to more advanced stages of the recovery process by day 10 post infection when tissues were harvested, most of the innate immune functions in the group treated with FPI and Ibuprofen for the longest time (group 6) were already reduced when compared to the

placebo. But the highest level of difference was observed between group 6 and group 2 as they were on the opposite sides of the spectrum of viral loads as measured by viral shedding. Building upon the knowledge base this study has generated, future studies in which tissue samples are acquired earlier in the disease process could be designed to further demonstrate the mechanisms by which treatment with a COX inhibitor with and without an anti-viral agent can alter BRSV pathogenesis.

## Supporting information

**S1 Fig. Multidimensional Scaling (MDS) plots in LL versus LN differential expression analysis.** When data sets were compared by lung tissue sample type; LL appeared on the area that was distinct from the area of LN.
(TIF)

**S2 Fig. Cluster of functionally connected GO terms in LN and their associated genes in the comparison of treatment group 6 vs 2.** Differential gene expression analysis in LN across treatment groups showed DE genes in treatment group 6 versus 2. Gene ontology enrichment analysis of these genes (p<0.05) was conducted and results were visualized as a map of predicted connections among significantly enriched GO terms and related genes.
(TIF)

**S1 File. Differentially expressed genes in comparison of lung tissue with lesion and lung tissue without lesion in groups 1–6.**
(XLSX)

**S2 File. Lists of differentially expressed genes that were unique for the certain treatment groups.**
(XLSX)

**S3 File. Result of GO enrichment analysis of differentially expressed genes in the comparison of lung lesions of groups 6 and 2.**
(XLSX)

**S4 File. Differential gene expression in lung lesions in group 6 compared with lung lesions in group 2.**
(XLSX)

**S5 File. Differential gene expression in lung tissue without lesion in group 6 compared with lung with no lesion in group 2.**
(XLSX)

**S6 File. Differential gene expression data viperin, prostaglandin synthases and FC epsilon signaling.**
(XLSX)

## Acknowledgments

The sequencing was carried out at the DNA Technologies and Expression Analysis Cores at the UC Davis Genome Center. The authors would like to thank specially Lutz Froenicke and Siranoosh Ashtari for their help with sequencing. Special thanks to Blythe Durbin-Johnson, Monica Britton and Matt Settles at UC Davis Bioinformatics Core for their assistance in handling and processing the sequencing data.

The antiviral FPI was generously donated by Gilead, Inc.

## Author Contributions

**Conceptualization:** Paul Walsh, Laurel J. Gershwin.

**Data curation:** Maxim Lebedev, Francisco R. Carvallo Chaigneau.

**Formal analysis:** Maxim Lebedev, Francisco R. Carvallo Chaigneau.

**Funding acquisition:** Paul Walsh, Laurel J. Gershwin.

**Investigation:** Maxim Lebedev, Heather A. McEligot, Victoria N. Mutua, Laurel J. Gershwin.

**Methodology:** Laurel J. Gershwin.

**Project administration:** Laurel J. Gershwin.

**Resources:** Maxim Lebedev, Heather A. McEligot, Victoria N. Mutua.

**Supervision:** Laurel J. Gershwin.

**Visualization:** Maxim Lebedev.

**Writing – original draft:** Maxim Lebedev.

**Writing – review & editing:** Victoria N. Mutua, Paul Walsh, Francisco R. Carvallo Chaigneau, Laurel J. Gershwin.

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
