## [Decision Letter · Decision Letter 0]

28 Oct 2020

PONE-D-20-30396

Analysis of lung transcriptome in calves infected with Bovine Respiratory Syncytial Virus and treated with antiviral and/or cyclooxygenase inhibitor

PLOS ONE

Dear Dr. Gershwin,

Thank you for submitting your manuscript to PLOS ONE. After careful consideration, we feel that it has merit but does not fully meet PLOS ONE’s publication criteria as it currently stands. Therefore, we invite you to submit a revised version of the manuscript that addresses the points raised during the review process.

 As cited by Reviewer-2, the authors should include additional data to support their claim on the genome-wide transcriptome analysis. It is important to relate the disease pathology (lung lesions) or no pathology (no lesions) to corresponding transcriptome profile. This is needed to understand the relevance of the networks/pathways/genes that were reported to be differentially regulated in this article. In addition, as noted by Reviewer-1, the authors should re-organize the write-up to eliminate repetition of their results in the discussion section, and instead, highlighting what are the unique characteristics of their study, compared to published literature. 

We look forward to receiving your revised manuscript.

Kind regards,

Selvakumar Subbian, Ph.D.

Academic Editor

PLOS ONE

Journal Requirements:

2. We note that you are reporting an analysis of a microarray, next-generation sequencing, or deep sequencing data set. PLOS requires that authors comply with field-specific standards for preparation, recording, and deposition of data in repositories appropriate to their field. Please upload these data to a stable, public repository (such as ArrayExpress, Gene Expression Omnibus (GEO), DNA Data Bank of Japan (DDBJ), NCBI GenBank, NCBI Sequence Read Archive, or EMBL Nucleotide Sequence Database (ENA)). In your revised cover letter, please provide the relevant accession numbers that may be used to access these data. For a full list of recommended repositories, see http://journals.plos.org/plosone/s/data-availability#loc-omics or http://journals.plos.org/plosone/s/data-availability#loc-sequencing

Reviewers' comments:

Reviewer's Responses to Questions

**Comments to the Author**

1. Is the manuscript technically sound, and do the data support the conclusions?

Reviewer #1: Yes

Reviewer #2: Yes

2. Has the statistical analysis been performed appropriately and rigorously? 

Reviewer #1: Yes

Reviewer #2: I Don't Know

3. Have the authors made all data underlying the findings in their manuscript fully available?

Reviewer #1: Yes

Reviewer #2: Yes

4. Is the manuscript presented in an intelligible fashion and written in standard English?

Reviewer #1: Yes

Reviewer #2: Yes

5. Review Comments to the Author

Reviewer #1: In the present study, the authors have presented lung transcriptomics of bovine lung upon BRSV infection. They further analyzed the effect of antiviral treatment regimen. In my opinion, it is a nice piece of work. However, I found redundant information have been provided at places. The discussion section can be presented precisely by removing the details of their findings, which have already been discussed in results section. Also, they should give precise information about any gene or physiological pathway while discussing.

Reviewer #2: Authors carried out differential gene expression analysis in Bovine Respiratory Syncytial virus (BRSV) infected lung tissues (LL vs LN) and shown that combined treatment with FPI and Ibuprofen, made the most difference in gene expression patterns in comparison with especially in pathways related to the innate and adaptive immune response in both LL and LN. Authors suggest that FPI in combination with ibuprofen would provide the best therapeutic intervention against BRSV infection. The study result is very interesting with future perspective but lack a lot of basic information. The present study has to address following questions.

Major comments:

Author investigated differential gene expression and pathway analysis between lung lesion (LL) and No lung lesion (LL) of different treatment groups. However, to understand the real time effect of your treatment, pathological and histo-pathological data must be included. Authors did not provide viral load in the lung as well as nasal swab in different group. Quantification of viral load will help to better interpret the results from gene expression and pathway analysis regarding to their phenotypical effect such as progression of disease.

Authors should provide pathological score to better understand how the treatment contributed to the resolution of disease/pathogenesis.

There is no histopathological data were provided, which help to determine clinical status of disease and also to understand the beneficial effect of treatment.

Authors analyzed differentially expressed gene and differentially regulated pathways between LL vs LN among treatment groups. It is interested to study the gene expression when the data analyzed between LL vs LL and LN vs LN among the treatment groups. Did authors carry out this analysis?

It will be better to estimate the production of PGE2, thromboxane and other pro-inflammatory cytokine/chemokine level to study the efficiency of treatment for control viral load as well as inflammation.

6. PLOS authors have the option to publish the peer review history of their article (what does this mean?). If published, this will include your full peer review and any attached files.

Reviewer #1: No

Reviewer #2: No

---

## [Author Response · Author response to Decision Letter 0]

30 Dec 2020

We have uploaded a separate response to reviewers. It is added here as well. 

Reviewer #1: In the present study, the authors have presented lung transcriptomics of bovine lung upon BRSV infection. They further analyzed the effect of antiviral treatment regimen. In my opinion, it is a nice piece of work. However, I found redundant information have been provided at places. The discussion section can be presented precisely by removing the details of their findings, which have already been discussed in results section. Also, they should give precise information about any gene or physiological pathway while discussing.

Thank you for this comment. We have reduced the redundancy in the discussion section and have retained only those comments required to facilitate the discussion. We have also altered the discussion and included some comparisons with the only other published paper examining the transcriptome of the lung in BRSV infected cattle. This comparison highlights some of the most important findings and helps to demonstrate the effects of treatments. These new additions are presented in lines 559-571 and 752-755.

We believe that we have identified most of the pathways and have attempted to clarify those that are not as obvious. Some pathways we encounter in the KEGG analysis are quite straightforward, such as KEGG:04610 complement and coagulation cascades. Others are less so, such as KEGG:05340 primary immunodeficiency. In this latter case we have attempted to explain to the reader that this means that there are some genes identified in our analysis that are shared with those in this pathway. For these analyses we did not always obtain the information regarding the exact genes involved, but where this data is available we have noted it.

Reviewer #2: Authors carried out differential gene expression analysis in Bovine Respiratory Syncytial virus (BRSV) infected lung tissues (LL vs LN) and shown that combined treatment with FPI and Ibuprofen, made the most difference in gene expression patterns in comparison with especially in pathways related to the innate and adaptive immune response in both LL and LN. Authors suggest that FPI in combination with ibuprofen would provide the best therapeutic intervention against BRSV infection. The study result is very interesting with future perspective but lack a lot of basic information. The present study has to address following questions.

Major comments:

Author investigated differential gene expression and pathway analysis between lung lesion (LL) and No lung lesion (LN) of different treatment groups. However, to understand the real time effect of your treatment, pathological and histo-pathological data must be included. Authors did not provide viral load in the lung as well as nasal swab in different group. Quantification of viral load will help to better interpret the results from gene expression and pathway analysis regarding to their phenotypical effect such as progression of disease.

We have provided the viral load as determined by qRT-PCR from daily nasal swabs beginning on day 0 prior to infection through day 10 when necropsy was performed. With this infection model, which we have previously performed and reported for other studies, the maximum viral load in the lung tissue occurs on or around day 7 post infection. By day 10, as demonstrated in other studies by both viral isolation attempts and immunoperoxidase staining for the virus, there is often little or no virus still present in the tissue. For this reason we did not attempt to quantitate virus in the lung tissue on day 10 of necropsy. The viral shedding data has been reported in reference 19 along with clinical data. A summary of this data has been included in the revision (lines 535-539).

Authors should provide pathological score to better understand how the treatment contributed to the resolution of disease/pathogenesis.

There is no histopathological data were provided, which help to determine clinical status of disease and also to understand the beneficial effect of treatment.

We have performed a very comprehensive analysis of the histopathology in this experiment. Six lung compartments (pleura, alveolus, septum, interstitium, bronchiole, and bronchus) were evaluated on H & E slides from each calf in the experiment by a board certified veterinary pathologist (FC). The evaluations included the scoring of specific histological lesions in each of these areas (from none, mild, moderate and severe). Statistical analysis was performed on this data. In addition, gross pathology consisting of determination of mean percentages of consolidation were computed and analyzed statistically. This information has been written up and is being submitted as a separate manuscript. It includes multiple microphotographs, figures and tables. It was our feeling that this very interesting and somewhat voluminous data required a separate manuscript. However, we totally understand why some part of the pathology should be included in the transcriptome manuscript. We have therefore included in the revision a brief summary of the pathology including mean percentages of lung consolidation in each group; we have added our pathologist (FC) as an author on this manuscript: see lines 541-553)

Authors analyzed differentially expressed gene and differentially regulated pathways between LL vs LN among treatment groups. It is interested to study the gene expression when the data analyzed between LL vs LL and LN vs LN among the treatment groups. Did authors carry out this analysis?

Yes, the LL and LN tissues within each group were analyzed and compared between groups. Differences and similarities were reported for those genes/pathways that were significantly involved. We have added some sub-titles to the discussion to make it easier to follow which comparisons are being made. Please see lines 812-860 and the section starting on line 862. 

It will be better to estimate the production of PGE2, thromboxane and other pro-inflammatory cytokine/chemokine level to study the efficiency of treatment for control viral load as well as inflammation.

We agree that the metabolomics in this experiment are a critical aspect for analysis. We have collected samples for metabolomics analysis and that work has been completed using mass spec. We are currently in the process of analysis of this mass spec. data and will be developing a manuscript specifically targeting this aspect of the work. It is our hope that the transcriptome manuscript and the pathology manuscript can then be cited in the metabolomics manuscript thereby providing a way for a reader to completely understand the total picture that emerges from this very large and lengthy experiment.

---

## [Decision Letter · Decision Letter 1]

25 Jan 2021

Analysis of lung transcriptome in calves infected with Bovine Respiratory Syncytial Virus and treated with antiviral and/or cyclooxygenase inhibitor

PONE-D-20-30396R1

Dear Dr. Gershwin,

We’re pleased to inform you that your manuscript has been judged scientifically suitable for publication and will be formally accepted for publication once it meets all outstanding technical requirements.

Kind regards,

Selvakumar Subbian, Ph.D.

Academic Editor

PLOS ONE

Additional Editor Comments (optional):

Reviewers' comments:

Reviewer's Responses to Questions

**Comments to the Author**

1. If the authors have adequately addressed your comments raised in a previous round of review and you feel that this manuscript is now acceptable for publication, you may indicate that here to bypass the “Comments to the Author” section, enter your conflict of interest statement in the “Confidential to Editor” section, and submit your "Accept" recommendation.

Reviewer #1: All comments have been addressed

Reviewer #2: All comments have been addressed

2. Is the manuscript technically sound, and do the data support the conclusions?

Reviewer #1: Yes

Reviewer #2: Yes

3. Has the statistical analysis been performed appropriately and rigorously? 

Reviewer #1: Yes

Reviewer #2: I Don't Know

4. Have the authors made all data underlying the findings in their manuscript fully available?

Reviewer #1: Yes

Reviewer #2: Yes

5. Is the manuscript presented in an intelligible fashion and written in standard English?

Reviewer #1: Yes

Reviewer #2: Yes

6. Review Comments to the Author

Reviewer #1: (No Response)

Reviewer #2: Authors tried to address all the comments and revised the manuscript. However, authors mentioned that some of the data asked by the reviewer will be published separate paper.

7. PLOS authors have the option to publish the peer review history of their article (what does this mean?). If published, this will include your full peer review and any attached files.

Reviewer #1: No

Reviewer #2: No

---

## [Editor Report · Acceptance letter]

3 Feb 2021

PONE-D-20-30396R1 

Analysis of lung transcriptome in calves infected with Bovine Respiratory Syncytial Virus and treated with antiviral and/or cyclooxygenase inhibitor 

Dear Dr. Gershwin:

I'm pleased to inform you that your manuscript has been deemed suitable for publication in PLOS ONE. Congratulations! Your manuscript is now with our production department. 

Kind regards, 

on behalf of

Dr. Selvakumar Subbian 

Academic Editor

PLOS ONE